# New Directions in Metal Phosphonate and Phosphinate Chemistry

**Stephen J.I. Shearan [1]**, **Norbert Stock [2]**, **Franziska Emmerling [3]**, **Jan Demel [4]**,
**Paul A. Wright [5]**, **Konstantinos D. Demadis [6]**, **Maria Vassaki [6]**, **Ferdinando Costantino [7]**,
**Riccardo Vivani [8]**, **Sébastien Sallard [9]**, **Inés Ruiz Salcedo [10]**, **Aurelio Cabeza [10]** and
**Marco Taddei [1,\*]**

[1]  Energy Safety Research Institute, Swansea University, Fabian Way, Swansea SA1 8EN, UK;
    940872@swansea.ac.uk
[2]  Institute of Inorganic Chemistry, Christian-Albrechts-University, Max-Eyth-Str. 2, 24118 Kiel, Germany;
    stock@ac.uni-kiel.de
[3]  Federal Institute for Materials Research and Testing (BAM), Richard-Willstaetter-Str. 11, 12489 Berlin,
    Germany; franziska.emmerling@bam.de
[4]  Institute of Inorganic Chemistry of the Czech Academy of Sciences, Husinec-Řež 1001, 250 68 Řež,
    Czech Republic; demel@iic.cas.cz
[5]  EaStCHEM School of Chemistry, University of St Andrews, Purdie Building, North Haugh, St Andrews
    KY16 9ST, UK; paw2@st-andrews.ac.uk
[6]  Crystal Engineering, Growth, and Design Laboratory, Department of Chemistry, University of Crete, Crete
    GR-71003 Heraklion, Greece; demadis@uoc.gr (K.D.D.); vassakimar@gmail.com (M.V.)
[7]  Department of Chemistry, Biology and Biotechnologies, Via Elce di Sotto n. 8, 06123 Perugia, Italy;
    ferdinando.costantino@unipg.it
[8]  Department of Pharmaceutical Sciences, University of Perugia, Via del Liceo 1, 06123 Perugia, Italy;
    riccardo.vivani@unipg.it
[9]  Flemish Institute for Technological Research—VITO, Sustainable Materials Department, Boeretang 200,
    2400 Mol, Belgium; sebastien.sallard@vito.be
[10]  Depto. Química Inorgánica, Cristalografía y Mineralogía, Campus de Teatinos s/n, Universidad de Málaga,
    29071 Málaga, Spain; inesrs@uma.es (I.R.S.); aurelio@uma.es (A.C.)
[\*]  Correspondence: marco.taddei@swansea.ac.uk; Tel.: +44-(0)1792-606230

**Abstract:** In September 2018, the First European Workshop on Metal Phosphonates Chemistry brought together some prominent researchers in the field of metal phosphonates and phosphinates with the aim of discussing past and current research efforts and identifying future directions. The scope of this perspective article is to provide a critical overview of the topics discussed during the workshop, which are divided into two main areas: synthesis and characterisation, and applications. In terms of synthetic methods, there has been a push towards cleaner and more efficient approaches. This has led to the introduction of high-throughput synthesis and mechanochemical synthesis. The recent success of metal–organic frameworks has also promoted renewed interest in the synthesis of porous metal phosphonates and phosphinates. Regarding characterisation, the main advances are the development of electron diffraction as a tool for crystal structure determination and the deployment of in situ characterisation techniques, which have allowed for a better understanding of reaction pathways. In terms of applications, metal phosphonates have been found to be suitable materials for several purposes: they have been employed as heterogeneous catalysts for the synthesis of fine chemicals, as solid sorbents for gas separation, notably $CO_2$ capture, as materials for electrochemical devices, such as fuel cells and rechargeable batteries, and as matrices for drug delivery.

**Keywords:** metal phosphonates and phosphinates; layered materials; metal–organic frameworks; synthesis; X-ray and electron diffraction; in situ characterisation; heterogeneous catalysis; gas sorption/separation; proton conduction; rechargeable batteries; drug delivery

## 1. Introduction

Metal phosphonates (MPs) are a class of inorganic–organic hybrid polymeric materials built by the coordination of phosphonate ligands to metal ions, forming extended structures of various dimensionalities [1]. The field of MPs chemistry has seen steady growth over the last few decades, which has been driven by the interest for applications in areas such as ion exchange [2], intercalation chemistry [3–5], proton conduction [6], catalysis [7], and others. This is principally due to the exceptional chemical and thermal stability and high insolubility of these materials in many solvents, which can be attributed to the hard character of the phosphonate oxygen atoms and their high coordination affinity for metal atoms. MPs chemistry turned 40 years old in 2018. In September 2018, the First European Workshop on Metal Phosphonates Chemistry was held at the Energy Safety Research Institute of Swansea University and attended by several researchers in the field of MPs based in Europe. The idea behind the event was to bring these researchers together for the first time and create a forum for discussion about the state of the art in the field and promote future collaborations. This article is intended to highlight some of the most promising new avenues of research discussed during the First European Workshop on Metal Phosphonates Chemistry. As such, we are not seeking to provide an exhaustive review of the recent progresses in the field, which have been covered by several other publications over the last few years [6,8–14], including a comprehensive book published in 2011 [1].

Since the field of MPs has come a long way, this article starts off by looking into the past and retracing some of the most important stages of early development, with a focus on structural chemistry. The third section is devoted to synthesis and characterisation methods, placing emphasis on innovative approaches such as high-throughput synthesis for the discovery of new compounds, mechanochemical synthesis, strategies to form porous frameworks, structure solution from electron diffraction, and in situ characterisation methods. The fourth section focuses on novel applications of MPs, including gas sorption/separation, catalysis, electrochemical devices (fuel cells and rechargeable batteries), and drug delivery. Finally, we take a look at what the future of the field might hold, trying to identify the most promising new directions for innovative research.

## 2. Historical Landmarks

The initial interest in the field stemmed from the work by Clearfield et al. in the field of tetravalent metal phosphates, which have been known for their ion exchange properties since the 1950s. The key step forward made by Clearfield was the determination of the crystal structure of $\alpha$-zirconium bis(monohydrogen orthophosphate) monohydrate [$Zr(HPO_4)_2 \cdot H_2O$, hereafter $\alpha$-ZrP] [15] (Figure 1a,b) from single-crystal X-ray diffraction (SCXRD) data in 1968. $\alpha$-ZrP has a layered structure constituted of Zr atoms octahedrally coordinated by tridentate monohydrogenphosphate groups, thus leaving free –OH groups pointing towards the interlayer space and hydrogen bonding to water molecules accommodated between the layers. The atomic level understanding of the structure of $\alpha$-ZrP triggered intense research that aimed at taking advantage of the acidic protons on the surface of the layers, especially for ion exchange and intercalation purposes.

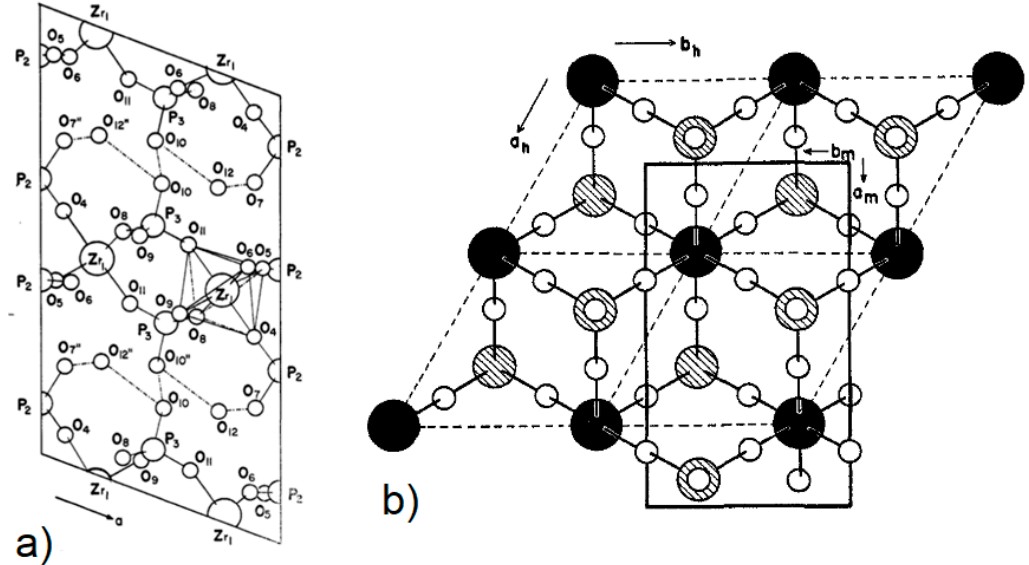

**Figure 1.** Crystal structure of α-ZrP viewed along the *b*-axis (**a**) and perpendicular to the layer (**b**). Adapted with permission from reference. [15]. Copyright 1969, American Chemical Society.

It was Alberti and Costantino et al. in 1978 [16] that would inaugurate the field of MPs. They prepared three zirconium phosphonates: zirconium phenylphosphonate, $Zr(C_6H_5PO_3)_2$; hydroxymethylphosphonate, $Zr(HOCH_2PO_3)_2$; and ethylphosphate, $Zr(C_2H_5OPO_3)_2$. Due to the very high insolubility of these compounds, they were unable to obtain suitable single crystals for solving the structures by SCXRD. Given the similar method of preparation to that of α-ZrP and based on the powder X-ray diffraction (PXRD) patterns dominated by intense basal peaks at low angles, the authors hypothesised that these compounds would give comparable layered structures to α-ZrP, where the –OH groups in the interlayer are substituted by the organic moieties, which determine the interlayer distance (Figure 2). In the following years, a number of similar compounds were prepared, based on both monophosphonates and diphosphonates, obtaining analogous layered or pillared-layered structures, respectively [17,18].

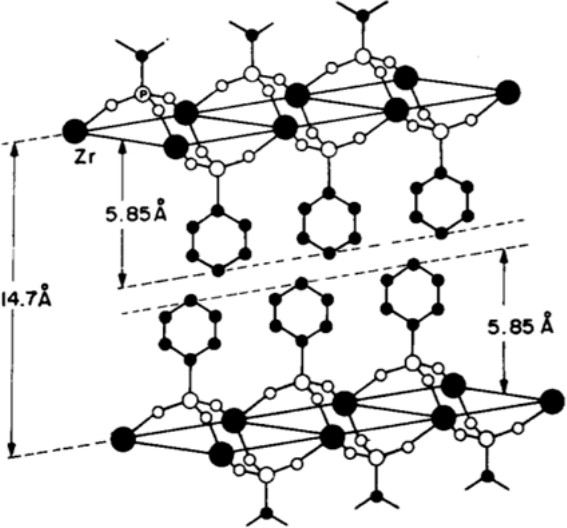

**Figure 2.** Idealised structural model of $Zr(C_6H_5PO_3)_2$. Reprinted with permission from reference [16]. Copyright 1978, Elsevier Inc.

The initial MP compounds were based on tetravalent metals, especially zirconium, but the late 1970s and 1980s brought about several structures based on divalent metal ions. Early work by Cunningham

et al. [19] looked at divalent metal phenylphosphonates and phenylarsonates, M($C_6H_5PO_3$) and (M($C_6H_5AsO_3$), with $M^{2+}$ = Mg, Mn Fe, Co, Ni, Cu, Zn, and Cd. They found that most of the synthetic processes were straightforward, and metal phenylphosphonates could be obtained by a simple reaction with the chloride or sulphate metal salts, with the exception of magnesium and iron. Thanks to the lower insolubility of these compounds, compared to tetravalent MPs, single crystals could be grown. Towards the end of the 1980s, a number of key papers detailing the crystal structures of various divalent MPs appeared [20–22]. All of these compounds featured structures based on layers built by the connection of metal atoms and phosphonate groups, with the organic moiety accommodated in the interlayer space. Figure 3 shows the layered crystal structure of Mn($C_6H_5PO_3$)·$H_2O$, which is just one of the examples provided by Cao et al. (1988) [21], where a number of structures based on divalent metals were described, based on SCXRD data, including: Mn, Mg, Ca, Cd, and Zn. Moving into the 1990s, MP structures expanded throughout the periodic table, covering a range of transition metals, all of the lanthanide series, and more than half of the s-block elements [1].

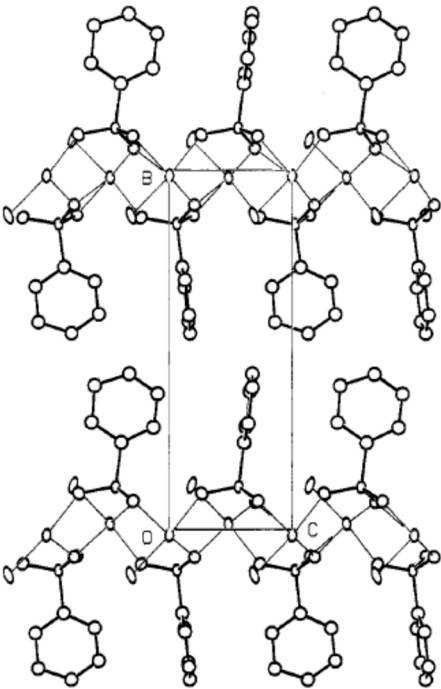

**Figure 3.** Structure of Mn($O_3PC_6H_5$)·$H_2O$ viewed down the *a*-axis. Reprinted with permission from reference [21]. Copyright 1988, American Chemical Society.

While this expansion across the periodic table was taking place, there was also significant progress made with solving the crystal structures from PXRD data. This was of great importance for the field of MPs, since it has often proven difficult to obtain suitable single crystals for SCXRD, especially when metals with high oxidation states were involved. An example that has already been noted comes from the pioneering work of Alberti and Costantino et al. [16], who were unable to obtain large crystals for Zr($C_6H_5PO_3$)$_2$. The structure was eventually solved by Poojary et al. in 1993 [23] from PXRD data, using a combination of modelling techniques and Patterson methods, and refined by Rietveld methods, confirming that the assumption made by Alberti and Costantino et al. 15 years before was indeed correct and the structure of Zr phenylphosphonate was based on the same layered arrangement of α-ZrP (Figure 4a,b). With the development of more sophisticated laboratory powder X-ray diffractometers, the increased accessibility of synchrotron sources, and the availability of more powerful crystallographic software, structural solutions from PXRD have progressively become a workhorse for researchers working with MPs.

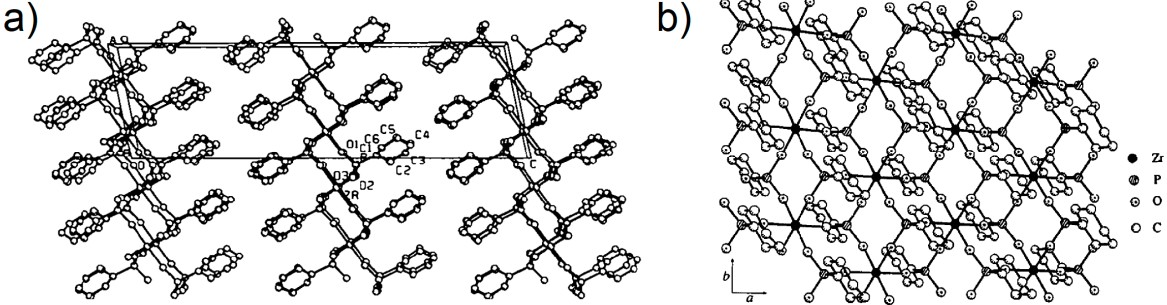

**Figure 4.** (**a**) Crystal structure of Zr(C$_6$H$_5$PO$_3$)$_2$ viewed along the *b*-axis, and (**b**) and perpendicular to the layer. Reprinted with permission from reference [23]. Copyright 1993, International Union of Crystallography.

While the overwhelming majority of MPs described in the first 15 years of research displayed layered structures, the early to mid-1990s witnessed the discovery of several open framework MPs [24–26]. The first examples were all based on the small ligand methylphosphonic acid, which afforded structures with channel-like arrangement, which was reminiscent of some zeolite frameworks, when combined with Cu, Zn, and Al (Figure 5a–c) [24–28]. The structural analogy with zeolites (and porous aluminium phosphates), which were at the time the most important class of crystalline and microporous materials, fuelled further investigation in the same direction [29]. This led to the discovery of more open framework materials, which were typically based on either monophosphonates or diphosphonates with short alkyl chains [30,31]. Increasing the length of the alkyl chains almost exclusively produced layered or pillared-layered structures, thus preventing the expansion of the channel size and generation of materials with higher porosity.

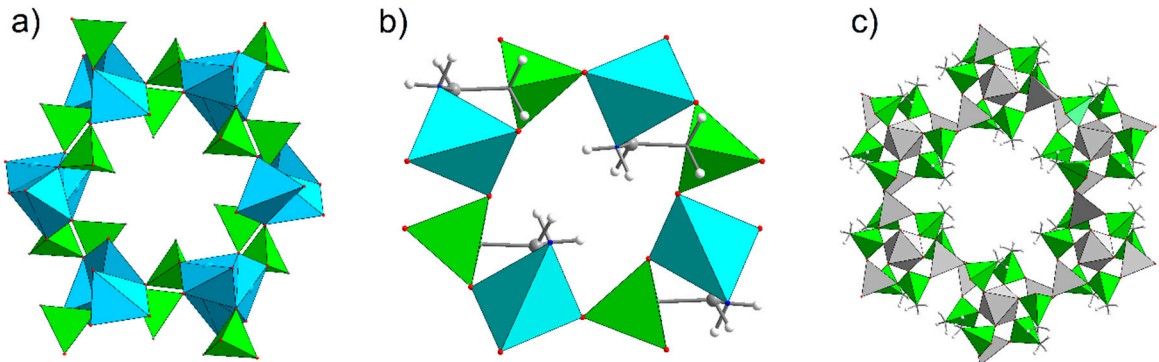

**Figure 5.** Crystal structures of the open framework metal phosphonates (MPs): (**a**) β-Cu(CH$_3$PO$_3$), (**b**) Zn(O$_3$PC$_2$H$_4$NH$_2$), and (**c**) α-Al$_2$(O$_3$PCH$_3$)$_2$. Colour code: light blue: copper, turquoise: zinc, light grey: aluminium, green: phosphorus, red: oxygen, blue: nitrogen, grey: carbon, white: hydrogen.

At the other end of the dimensionality scale, molecular MPs also started being the object of research in the second half of the 1990s, primarily for their potential as single molecule magnets and as model compounds for phosphate materials, especially those of group 13 elements (Figure 6) [9,32–36]. In order to overcome the strong tendency of MPs to polymerise, three strategies were devised to prevent the expansion of the structure: (1) using terminal ancillary ligands that can occupy coordination sites on the metal ions; (2) introducing sterically demanding groups on the phosphonic acids' backbone; (3) using a preformed cluster and performing controlled ligand exchange [9].

The reader interested in a more comprehensive account about the progress of the field until 2011 can refer to the book "Metal Phosphonate Chemistry: from Synthesis to Applications" [1], which also includes a chapter on the early history of MPs chemistry, authored by Abraham Clearfield [37].

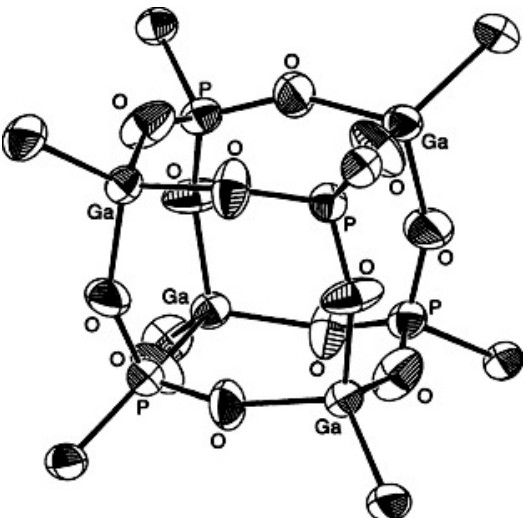

**Figure 6.** Crystal structure of the $Ga_4P_4O_{12}$ core found in $[^tBuGa(\mu_3–O_3PPh)]_4$. Reprinted with permission from reference [32]. Copyright 2003, Wiley-VCH Verlag GmbH & Co.

## 3. Synthesis and Characterisation of Metal Phosphonates

### 3.1. High-Throughput Methods

The synthesis of MPs is usually performed under hydrothermal or solvothermal conditions, with reaction times in the range of a few hours up to several days [38]. Identifying the proper conditions to obtain crystalline products involves the screening of reaction parameters such as pH, temperature, the concentration and molar ratios of reagents, and the amount of mineralisers or crystallisation modulators, which makes the discovery of novel MPs a time-consuming process. In addition, each "metal phosphonate" reaction system is unique, with its own idiosyncrasies. High-throughput (HT) methods have long been used to screen for ideal synthesis parameters, often leading to the accelerated discovery of novel compounds and further optimisation for increased yield [39] (Figure 7). This is also true for their application in MP synthesis, with the first reports of this approach falling in the early 2000s [40–44], and the methods are nowadays routinely used [45–47]. The interest in HT methods stems from their high efficiency, allowing for the systematic investigation of reaction parameters.

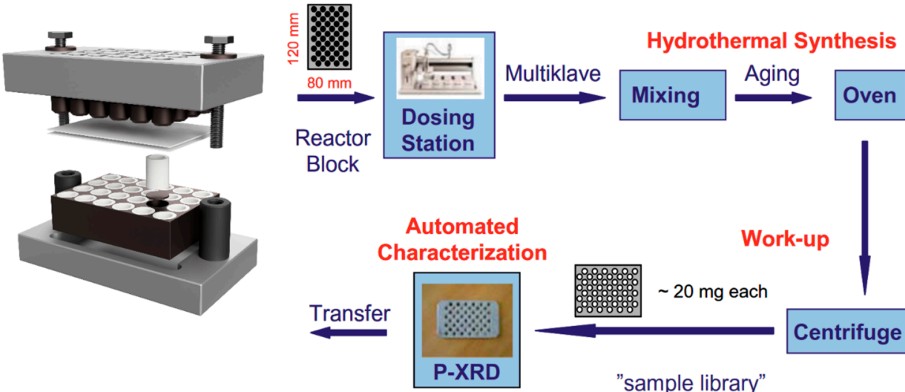

**Figure 7.** Typical workflow for a high-throughput (HT) synthesis experiment. Adapted with permission from reference [38]. Copyright 2009, Elsevier Inc.

Some of the earliest work on HT methods applied to MPs was carried out in 2004 [40], and looked at MPs based on the ligands $(H_2O_3PCH_2)_2N(CH_2)_4N(CH_2PO_3H_2)_2 \cdot 2H_2O$ ($H_8L^1 \cdot 2H_2O$), and $p\text{-}H_2O_3PCH_2C_6H_4COOH$ ($H_3L^2$). The investigation began with the design of two experiments to look at the system $Zn(NO_3)_2/H_3L^2/NaOH$. The first of the two varied the molar concentrations of the

three starting materials while keeping the water content the same (Figure 8). This led to the discovery of three single crystalline compounds $Zn(HO_3PCH_2C_6H_4COOH)_2$ **(1)**, $Zn(O_3PCH_2C_6H_4COOH)$ **(2)**, and $Zn_3(HO_3PCH_2C_6H_4COO)_2 \cdot 4H_2O$ **(3)**. Throughout the experiment, pH measurements were taken periodically, allowing the researchers to correlate the pH measurements of the reaction mixture with the dimensionality of the MP structure. Thus **(3)**, featuring a three-dimensional (3D) framework structure based on highly condensed inorganic units, was favoured at pH $\approx$ 6, whereas **(1)** (pH $\approx$ 0) and **(2)** (pH $\approx$ 1) displayed isolated columns and layered structures, respectively. The second experiment used the opposite parameters to the first, with the molar ratio for $Zn(NO_3)_2/H_3L^2/NaOH$ kept constant while adjusting the water content. This led to the discovery of a fourth phase, $Zn(HO_3PCH_2C_6H_4COOH)_2 \cdot 4H_2O$ **(4)**.

The researchers also carried out an investigation of the system $M^{2+}/H_8L^1 \cdot 2H_2O$ based on previous results [48]. In this case, various divalent ions were employed: M = Mg, Ca, Sr, Ba, Mn, Fe, Co, Ni, Zn, Cd, Sn, and Pb. As with the Zn-phosphonates, the investigation was split between two experiments, with the first taking the form of a discovery library for which the molar ratios for the starting materials are adjusted (e.g., 1:1, 1:4, 4:1, 2:3), while the water content was kept constant. Results showed that the microcrystalline products had formed for M = Mg, Ca, Mn, Fe, Co, Ni, Zn, and Cd, and that the compounds were isostructural. The second experiment showed that it was also possible to obtain single crystals, where M = Mg, Mn, Co, Ni, Cd, and Zn, by keeping the $M^{2+}:H_8L^1$ molar ratio at 1:1 and adjusting the water content between 98–99.83 mol % in order to tune the concentration of the starting materials. In total, 48 individual reactions were carried out without direct manipulation of each sample, while also exploring a large parameter space.

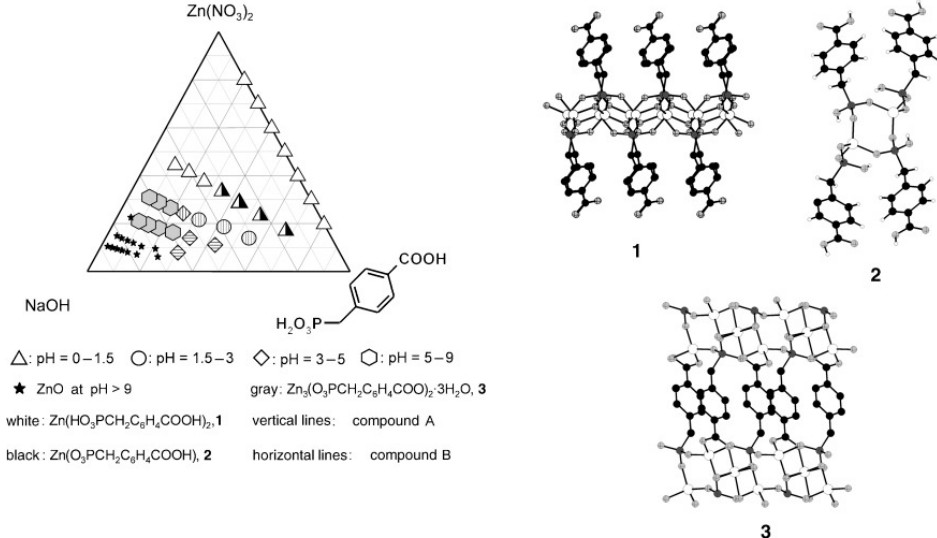

**Figure 8.** Crystallization diagram of observed phases in the $Zn(NO_3)_2/p\text{-}H_2O_3PCH_2C_6H_4COOH/NaOH$ system (left). Sections of the structures of $Zn(HO_3PCH_2C_6H_4COOH)_2$ **(1)**, $Zn(O_3PCH_2C_6H_4COOH)$ **(2)**, and $Zn_3(HO_3PCH_2C_6H_4COO)_2 \cdot 4H_2O$ **(3)** (right). Colour code: Zn: white, C: black, P: dark grey, O: light grey. Reprinted with permission from reference. [40]. Copyright 2004, Wiley-VCH Verlag GmbH & Co.

Subsequent publications explored the multifunctional metal carboxyphosphonates [41,43] via the method considered previously, successfully obtaining a range of structures based on Co and Mn. Over the years, several new MPs have been discovered thanks to the application of HT methods [45,49], whose efficacy was also proved for less conventional synthetic approaches, such as microwave-assisted synthesis [50] and ultrasonic synthesis [51].

One of the initial drawbacks identified for HT methods has been the inability to control the temperature for each individual reaction. Since temperature has shown to play such a profound role in

determining the product outcome and structures obtained, it would be worth having more control of this factor, allowing for a larger parameter space than is already available to be explored. The solution to the problem of temperature control was found in a thermocycler, or a polymerase chain reaction machine, which is usually found in the field of genetics [39]. This has allowed for the control of temperature for individual reaction chambers.

### 3.2. Mechanochemical Synthesis

Mechanochemistry has a long history extending back over 1000 years [52,53] and had not, until recently, received mainstream attention as a viable tool in chemical synthesis. However, over the last two decades, there has been a massive push towards green and sustainable chemistry, which is perhaps one of the main reasons why mechanochemistry has since received a lot more attention in a range of specialised fields. While there are still a lot of details to be ironed out with regards to the specific mechanisms that drive mechanochemical synthesis, the basic principle is clear. The input of mechanical energy, i.e., through grinding/milling drives the reaction between two or more solids to form a desired product, which is often in the presence of little or no solvent [54]. One of the key points that makes mechanochemistry more appealing is that it is possible for reactions to proceed via pathways that are not accessible through conventional methods [55].

Some of the most significant work on the mechanochemical synthesis of MPs has been carried out over the past four years. In 2016, a synthesis was reported using a vibration ball mill, in which cadmium acetate dihydrate and phenylphosphonic acid were combined inside a reaction vessel in various ratios (1:1, 1:2, and 1:4) and then milled for 15 min along with two 4.0-g stainless steel balls. One known and two novel cadmium phenylphosphonates were obtained: $Cd(O_3PPh)\cdot H_2O$ (Figure 9a,d), $Cd(HO_3PPh)_2$ (Figure 9b,e), and $Cd(HO_3PPh)_2(H_2O_3PPh)$ (Figure 9c,f) [56]. Each of the products obtained after milling were damp, which was caused by the release of water and acetic acid during the reaction, which in turn, means that each section of the synthesis was in fact liquid-assisted.

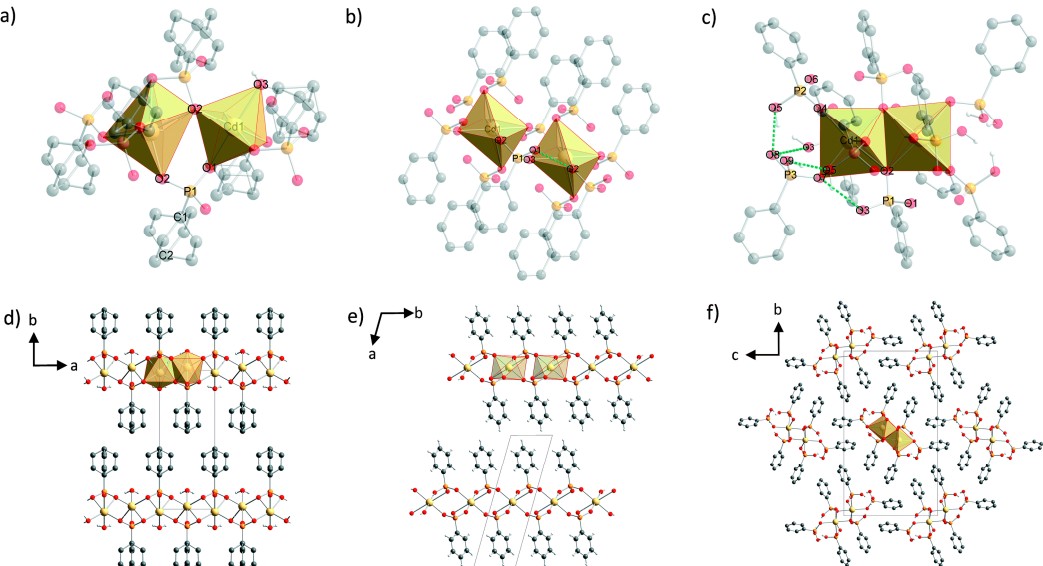

**Figure 9.** Molecular structure and coordination polyhedra for the $Cd^{2+}$ ions and hydrogen bonds (green dashed lines) of (**a**) $Cd(O_3PPh)\cdot H_2O$, (**b**) $Cd(HO_3PPh)_2$, (**c**) $Cd(HO_3PPh)_2(H_2O_3PPh)$, and the crystal structures of $Cd(O_3PPh)\cdot H_2O$, (**d**) viewed along the *c*-axis, (**e**) $Cd(HO_3PPh)_2$, viewed along the *c*-axis, and (**f**) $Cd(HO_3PPh)_2(H_2O_3PPh)$, viewed along the *a*-axis. Hydrogen atoms of the phenyl rings are omitted for clarity. Colour code: yellow: cadmium, orange: phosphorus, red: oxygen, grey: carbon, light grey: hydrogen. Adapted with permission from reference [56]. Copyright 2016, Royal Society of Chemistry.

In the same year, further work was carried out which looked at substituting cadmium with various other metals. The authors state that this is the first work that explores the mechanochemical synthesis of molecular MPs [57]. They present isomorphic structures of molecular MPs, $M(HO_3PPh)_2(H_2O_3PPh)_2(H_2O)_2$, where M = Mn, Co, and Ni (Figure 10a,b). Using the previously established procedure, the researchers were able to successfully synthesise three pure compounds after neat grinding for 15 min. They also carried out a liquid-assisted grinding (LAG) run for each of the compounds, finding that it had no effect on the data obtained when compared to the "dry" run. The conclusion was that this approach could be applied to the synthesis of other molecular MPs and would bring great benefits in terms of its simplicity and speed. There would also be opportunity for industrial applications due to the scalability, small environmental impact, and once again, the speed of synthesis.

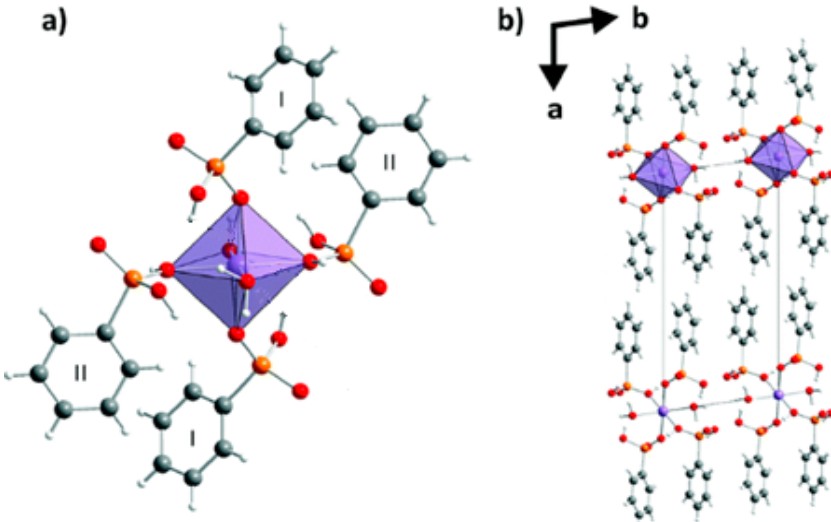

**Figure 10.** Crystal structure of $[Mn(HO_3PPh)_2-(H_2O_3PPh)_2(H_2O)_2]$ **(1),** with respective values for the isomorphic compounds based on Co **(2)** and Ni **(3)**. (**a**) Coordination sphere of the metal ion. (**b**) Structure of **1** shown along the *c*-axis (b). Hydrogen atoms are omitted for clarity. Colour code: purple: manganese, red: oxygen, orange: phosphorus, grey: carbon, light grey: hydrogen. Adapted with permission from reference. [57]. Copyright 2016, Royal Society of Chemistry.

More recent additions to the body of work looked at exploiting MPs with N-containing ligands. In 2017, the compound zinc N-(phosphonomethyl)glycinate $Zn(O_3PCH_2NH_2CH_2CO_2)\cdot H_2O$ was reported, displaying a 3D pillared-layered structure (Figure 11a–c) [58]. This was achieved through the LAG of zinc acetate dihydrate and N-(phosphonomethyl)glycine in a 1:1 ratio in the presence of water over 15 min. Through PXRD, it was shown that after just 30 s, the starting materials was almost completely consumed, and were completely absent after three minutes. Rietveld refinement proved that a pure compound is obtained. Further work on N-containing phosphonate ligands was carried out on manganese phosphonates [59], namely manganese mono(nitrilotrimethylphosphonate) (MnNP$_3$) and manganese bis(N-(carboxymethyl)-iminodi(methylenephosphonate)) $[Mn(NP_2AH)_2]$.

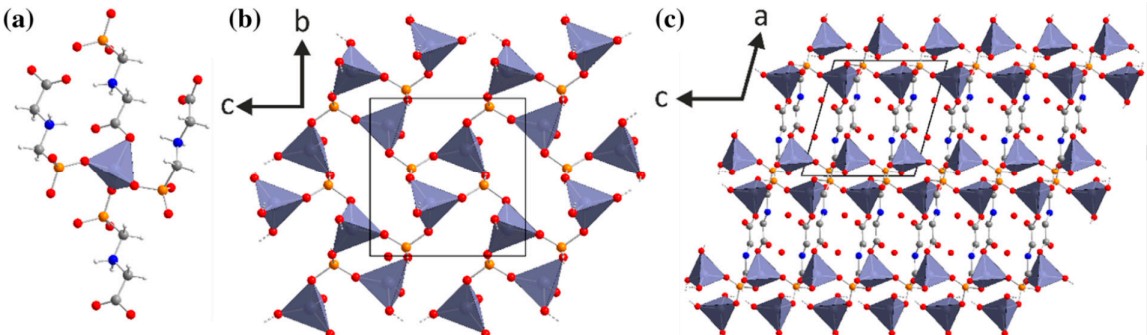

**Figure 11.** Structure of Zn(O$_3$PCH$_2$NH$_2$CH$_2$CO$_2$)·H$_2$O. (**a**) Coordination sphere of the Zn$^{2+}$ ion. (**b**) Structure of the layer formed by the ZnO$_4$-tetrahedra and the phosphono groups, shown along the *a*-axis. (**c**) Pillared structure of the framework shown along the *b*-axis (c). Hydrogen atoms are omitted for clarity. Colour code: Purple: zinc, red: oxygen, orange: phosphorus, blue: nitrogen, grey: carbon, and white: hydrogen. Adapted with permission from reference. [58]. Copyright 2017, Springer Science Business Media.

Numerous efforts have also been made to combine mechanochemical synthesis and in situ characterisation methods, which are discussed more in detail in Section 3.5, herein.

It is worth reiterating that mechanochemical synthesis provides a range of advantages over more conventional methods. It is a facile technique that provides results in relatively short periods of time, e.g., 15 min, and full conversion is often achieved. It is also worth considering the green status of the technique, owing to the requirement for little to no solvent. Then, it is clear that mechanochemical synthesis is an invaluable tool on any scale requirement, and could provide researchers with an alternative to conventional synthesis methods, yielding products that may have previously been inaccessible.

### 3.3. Porous Materials

The structures of most of the early MPs were two-dimensionally (2D) layered [15,16], whereby organic molecules would coordinate to a central metal layer via the phosphonate group. Then, the organic functionalities would interlace with other metal–organic layers, creating a dense, multi-layered system. As discussed in Section 2, some early examples of open framework MPs were reported in the 1990s which, with few exceptions [60], displayed pores too small to accommodate guest species larger than water, and whose porosity could not be probed by the adsorption of N$_2$ at 77 K [29]. In an almost parallel arc to open framework MPs, another class of porous inorganic–organic materials, i.e., metal–organic frameworks (MOFs), which are mainly based on carboxylic or N-heterocyclic ligands, started attracting growing interest from 1999 onwards [61–63]. Since then, MOFs have become one of the most intensely investigated classes of materials for applications such as gas separation and storage, catalysis, and drug delivery, owing to their large porosity, ease of functionalisation, and structural versatility [64,65]. Phosphonate-based MOFs represent a rather small fraction of the thousands of MOF structures reported over the last 20 years. This is mainly due to the strong tendency of MPs to form extended inorganic layers, coupled with the tendency of the phosphonate moiety to bridge metal ions. This poses significant synthetic challenges when the aim is to generate open frameworks. However, some interesting advances have been made recently in developing synthetic approaches that yield permanently porous MP materials [12–14,66].

One of the first examples of permanently porous MPs is MIL-91 [67]; this was based on the linker *N*,*N*′-piperazinediphosphonic acid, and was initially reported in 2006 by Serre et al. as two isoreticular analogues based on Al(III) and Ti(IV) (Figure 12a–c). The structures display monodimensional inorganic building units (IBUs) and channels running along the *b*-axis with an internal space of around 3.5 × 3.5 Å. A Langmuir surface area of around 500 m$^2$·g$^{-1}$, as well as a pore volume of 0.20 cm$^3$·g$^{-1}$,

were measured by $N_2$ adsorption at 77 K. The compounds were also shown to be reasonably stable up to 550 K, after which there is a significant breakdown of the organic groups, causing the structure to become X-ray amorphous. The synthesis of both MIL-91(Al) and MIL-91(Ti) was successfully scaled up to the 100-g scale with milder procedures than those originally reported, i.e., using reflux instead of hydrothermal conditions, reducing the synthesis time and avoiding the use of hydrofluoric acid as a mineraliser [68]. A later study identified some unusual adsorption properties of the structure towards $CO_2$, which will be examined in detail in Section 4.2 [69].

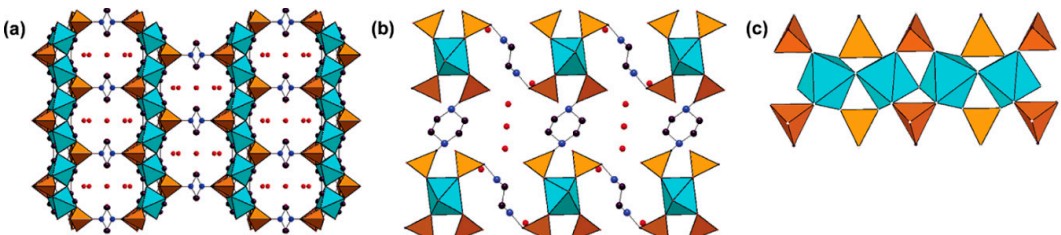

**Figure 12.** Crystal structure of MIL-91 viewed (**a**) along the *c*-axis and (**b**) the *b*-axis. For clarity, only one set of the two symmetry-equivalent (but half-occupied) sets of diphosphonate ligands connecting octahedral chains along *c* are shown. (**c**) A corner-sharing chain of $MO_6$ octahedra (c). Colour code: titanium: cyan, phosphorus: orange, oxygen: red, nitrogen: blue, carbon: purple. Adapted with permission from reference [67]. Copyright 2006, American Chemical Society.

The same ligand was also employed to prepare a series of isoreticular compounds (named STA-12) based on divalent metals, namely Ni, Co, Fe, and Mn, and featuring large hexagonal channels running along the *c*-axis (Figure 13) [70]. Interestingly, the STA-12 framework displays dehydration behaviour that is dependent on the metal atom: while STA-12(Ni) and STA-12(Co) are porous to $N_2$ when coordinated water molecules are removed from the metal ions in the IBUs, the Mn and Fe analogues are not [71]. This was attributed to the tilting of the piperazine rings and filling of the open coordination sites on Fe and Mn by the uncoordinated P–O group, which inhibits the uptake of $N_2$ probably by blocking access of the pores on the external surface. Using the linker *N,N*′-bispiperidinediphosphonic acid, an isoreticular analogue of STA-12, named STA-16, was obtained with Ni and Co as the metal atoms (Figure 13) [72]. Thanks to the longer linker, STA-16 features pores with 1.8-nm diameter, approaching the mesoporous regime, and a large pore volume of 0.68 $cm^3 \cdot g^{-1}$. So far, this remains a unique example of isoreticular expansion for porous MPs.

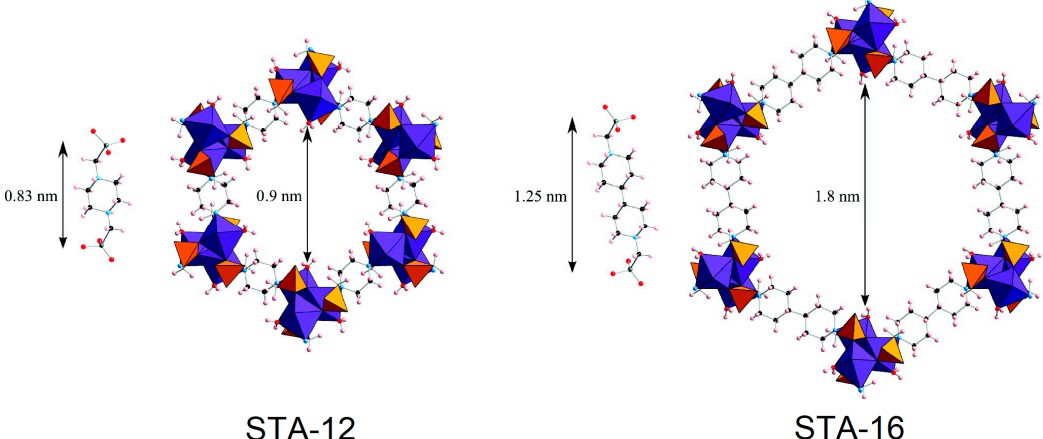

STA-12             STA-16

**Figure 13.** Crystal structures of (**left**) STA-12, based on *N,N*′-piperazinediphosphonic acid (**left**) and (**right**) STA-16, based on *N,N*′-bispiperidinediphosphonic acid (right). Colour code: cobalt, nickel: purple, phosphorus: orange, oxygen: red, nitrogen: light blue, carbon: black, hydrogen: white. Adapted with permission from reference [72]. Copyright 2011, American Chemical Society.

More recently, the use of rigid linkers with non-linear geometries was proven to be a viable strategy to induce the formation of porous structures [14]. Among them, tritopic linkers have been particularly investigated by several groups, and a recent review on the topic has been published, to which the interested reader is directed [73]. Besides tritopic linkers, tetratopic ones with either tetrahedral or square geometry have also proved successful in generating open framework compounds [74–77]. A notable example is the linker Ni-tetrakis(4-phosphonophenyl)porphyrin, which was combined with both divalent metals, namely, Mn, Co, Ni, and Cd [46,78], and tetravalent metals, namely Zr and Hf [47], to produce permanently porous compounds, termed CAU-29, CAU-36, and CAU-30, respectively (Figure 14a,b). HT methods were extensively employed to identify the best synthetic conditions to induce the formation of single crystals (in the case of divalent metals) and microcrystalline solids (in the case of tetravalent metals). Notably, the CAU-30 framework combines one of the most porous structures ever reported so far for MPs, featuring a specific surface area of almost 1000 $m^2 \cdot g^{-1}$ and a pore volume of 0.5 $cm^3 \cdot g^{-1}$, and exceptional stability, retaining its crystallinity up to 400 °C and pH 12.

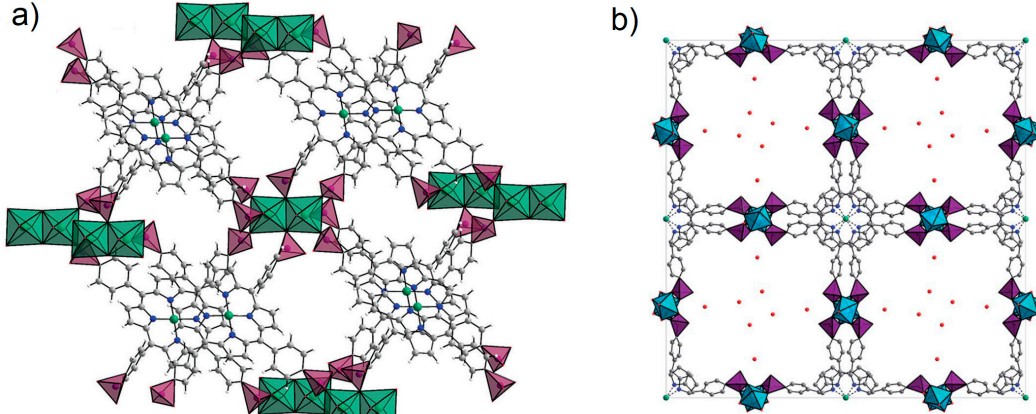

**Figure 14.** Crystal structures of (**a**) CAU-29 [Reprinted with permission from reference. [46]. Copyright 2018, Royal Society of Chemistry] and (**b**) CAU-30 [Reprinted with permission from reference [47]. Copyright 2018, Royal Society of Chemistry. Colour code: nickel: green, zirconium: light blue, phosphorus: purple, oxygen: red, nitrogen: dark blue, carbon: grey, hydrogen: white.

While the field of carboxylate-based MOFs has moved beyond the initial phase—which was dominated by crystal engineering—towards practical applications and large-scale deployment, progress in phosphonate-based MOFs has been slower, and it still remains challenging to identify the right combination of linker, metal, and synthetic conditions to generate permanently porous structures. This section has showed that the recently renewed interest in microporous MPs has already produced some notable results, which hold promise for further advances in the near future.

### 3.4. Phosphinic Acid-Based Materials

All previous discussion in this review has focused on phosphonic acid-based materials. However, a closely related class of compounds has also been explored: so-called phosphinic acid-based materials. Phosphinic acids are different from phosphonic acids, in that one of the –OH groups on phosphorous is replaced by a second organic moiety. This makes phosphinates less versatile in terms of coordination ability, but at the same time provides the possibility to exploit the nature of the organic moieties attached to the P atom. A comparison of the structure of phosphonic and phosphinic acids can be seen in Figure 15. Thus far, metal phosphinates have received less attention than phosphonates, and most of the reported structures are based on either diphenylphosphinic acid or diphosphinates with small organic groups, which are sometimes combined with auxiliary ligands [79,80]. Besides their similarity with phosphonic acids, phosphinic acids also share some features with carboxylic acids: they are monovalent groups with two oxygen atoms available to coordinate to metal species, although

the different hybridisations of the phosphinic P atom ($sp^3$) and the carboxylic C atom ($sp^2$) have obvious effects on the geometry of the two groups (Figure 15). Notably, phosphinic acids are more acidic than carboxylic acids, which is the reason why metal phosphinates have shown much better hydrolytic stability. As mentioned above, carboxylic acids have usually been the linkers of choice in "traditional" MOF chemistry. A few examples of microporous compounds based on phosphonate monoesters (Figure 15), which share some features with phosphinic acid linkers, have been reported recently [81–84]. On the other hand, until very recently, no permanently porous metal phosphinate MOFs were reported.

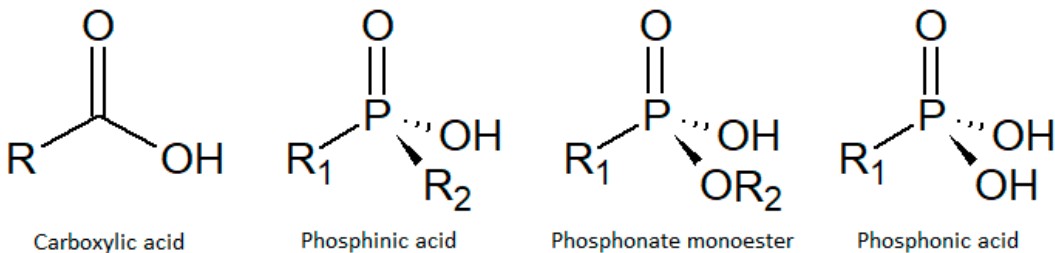

**Figure 15.** Molecular structures of carboxylic, phosphinic, and phosphonic acids.

In 2018, Hynek et al. [82] used the linker phenylene-1,4-bis(methylphosphinic acid), hereafter denoted PBPA(Me), which can be seen as an analogue of terephthalic acid, the most common linker in MOF chemistry. Three different compounds were obtained by the reaction of PBPA(Me) with Fe(III) salts in different conditions (Figure 16): ICR-1 (ICR stands for Institute of Inorganic Chemistry Řež), prepared in water, has a non-porous 3D framework, which is isoreticular to a previously reported Zr phosphonate [85]; ICR-2, prepared in ethanol, is instead porous, with hexagonal channels with diameters of 0.8 nm running along the *c*-axis and a Brunauer-Emmett-Teller (BET) surface area of 731 m$^2\cdot$g$^{-1}$; ICR-3, prepared in *N,N'*-dimethylformamide is a layered dense phase. ICR-2 was shown to be stable in a range of solvents, most notably water. The authors were also able to obtain a compound isoreticular to ICR-2, termed ICR-4, by employing the linker phenylene-1,4-bis(phenylphosphinic acid) [PBPA(Ph)]. Due to the larger size of the organic groups accommodated in the pores, ICR-4 has a much lower surface area (165 m$^2\cdot$g$^{-1}$) than ICR-2.

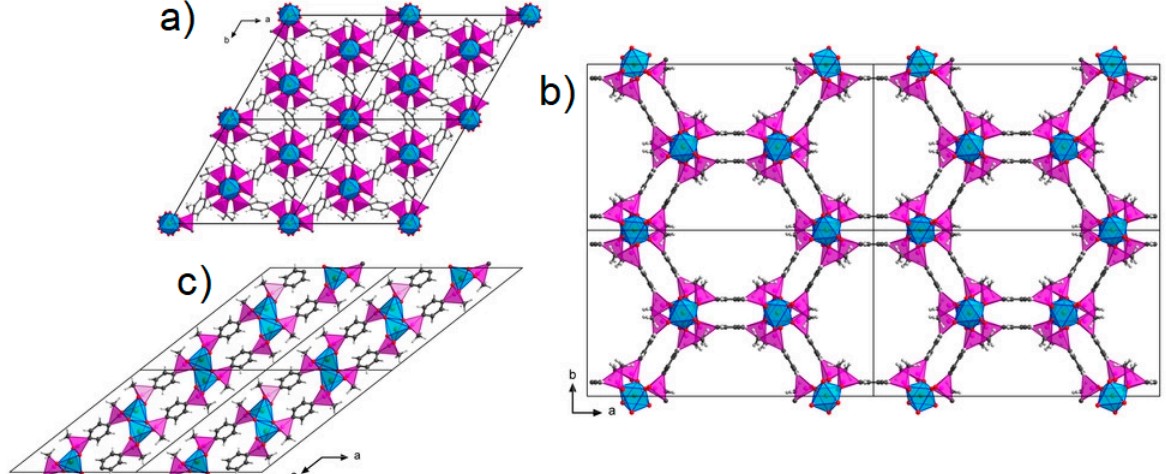

**Figure 16.** Crystal structures of (**a**) ICR-1, (**b**) ICR-2, and (**c**) ICR-3. Colour code: iron: blue, phosphorus: pink, oxygen: red, carbon: grey, hydrogen: white. Adapted with permission from reference. [82]. Copyright 2018, Wiley-VCH Verlag GmbH & Co.

The results reported by Hynek et al. demonstrate that phosphinic acids can also be suitable linkers for the construction of open framework structures, with the added advantage that the pendant organic

group can be modified and/or functionalised to impart specific properties to the material. This is not possible in the case of phosphonates and carboxylates; therefore, the use of phosphinic acids might open up a whole new range of opportunities in the field of MOFs.

### 3.5. Structure Determination by Electron Diffraction

2018 has seen a major breakthrough for the structural characterisation of microcrystalline and nanocrystalline compounds, mostly thanks to two major papers reporting on the use of electron diffraction (ED) data to solve the crystal structures of some small organic molecules [86,87]. Using ED, it is possible to obtain diffraction patterns that are comparable to those usually observed when performing SCXRD experiments from crystals of submicrometric size. The method is particularly attractive because it does not require exotic instrumentation and, in principle, any transmission electron microscope could be adapted for such a task, holding promise for a rapid deployment of the technique as a routine tool for structural characterisation.

Even before the two aforementioned papers appeared, ED had already been employed to determine the crystal structure of some MOFs [88], including some compounds mentioned earlier in this review: CAU-30, CAU-36, (Section 3.3) [47,78], ICR-1, ICR-2, and ICR-3 (Section 3.4) [82]. All these compounds were obtained as microcrystalline powders, whose crystallite size was below the micrometre range and not sufficiently high to allow structure solution from PXRD data (Figure 17). In the case of CAU-30, the structural model was obtained from automated electron diffraction tomography [89], whereas for the ICR compounds, micro ED [90] was employed. In order to minimise beam damage, the samples were cooled to 95 K (CAU-30) or 100 K (ICRs), and the beam was shifted on the crystals. For all the compounds, the structural model was then validated by classical Rietveld refinement of the PXRD patterns. Another notable example of the use of ED in MP chemistry is represented by CAU-36 [78], which was a Co-based porous compound obtained using the same Ni-tetrakis(4-phosphonophenyl)porphyrin linker employed to prepare CAU-29 and CAU-30. In the case of CAU-29, continuous rotation electron diffraction [91] was employed, allowing to precisely identify the location of the guest species (1,4-diazabicyclo[2.2.2]octane and solvent molecules) within the framework.

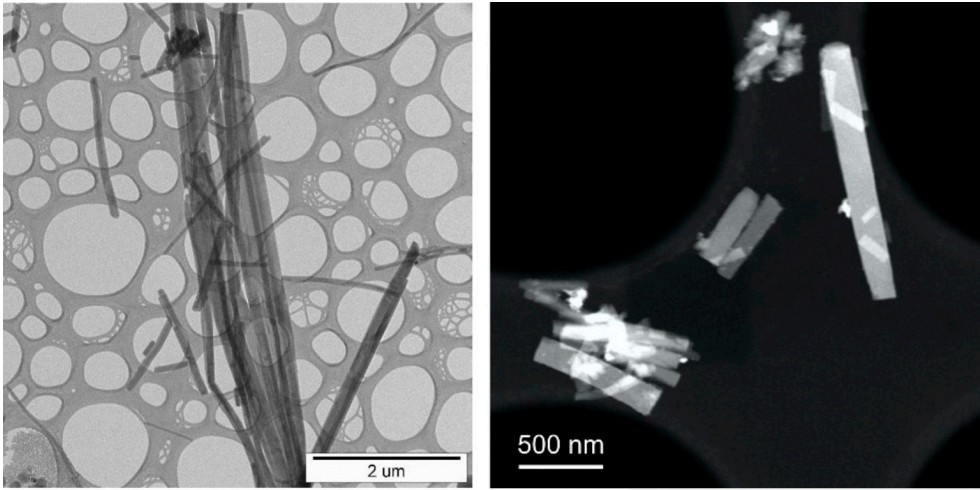

**Figure 17.** Transmission electron micrographs of ICR-2 (**left**) (Reprinted with permission from reference. [82]. Copyright 2018, Wiley-VCH Verlag GmbH and Co.) and CAU-30 (**right**) (Reprinted with permission from reference. [47]. Copyright 2018, Royal Society of Chemistry).

Given the difficulty in obtaining suitable crystals for SCXRD, especially when high oxidation state metals are employed, the structural determination of MPs has often been carried out from PXRD data. However, due to the limited amount of information contained in a PXRD pattern, some structures can require a large amount of time to be solved, if they can be solved at all. This is one of the main factors that have prevented the field of MPs from progressing at the pace of other classes of coordination

polymers, such as carboxylate-based MOFs. Therefore, the possibility of accessing ED as a tool for structural determination can be an absolute game changer in this sense.

### 3.6. In Situ Characterisation

In situ investigations have been widely used in materials chemistry to monitor synthetic processes, especially those involving the self-assembly of well-defined building blocks into crystalline compounds [92]. Besides simply following the products' formation, this approach is powerful in identifying the different phases and intermediates that form over the course of a reaction [93]. The application of in situ investigations can take many different forms and draw on a wide range of techniques, such as XRD, X-ray absorption spectroscopy, and vibrational spectroscopies [94]. The data obtained from in situ studies can be used to identify the different stages in a reaction and correlate this to the reaction parameters for further optimisation.

An early example of in situ investigations of MPs stems from Feyand at al. and the work carried out on the HT synthesis of lanthanide phosphonatobutanesulfonates, $Ln(O_3P-C_4H_8-SO_3)(H_2O)$, where Ln = La – Gd [95]. They managed to produce a series of isostructural compounds on which they carried out in situ analysis using energy-dispersive X-ray diffraction (EDXRD) during both conventional and microwave-assisted heating processes. A number of phases were identified before the formation of the final product (Figure 18). During the first five minutes, there was some background modulation, indicating that an amorphous side phase had formed in the initial part of the reaction, but after one minute, there was clear evidence for a crystalline intermediate. The formation of the product was observed at seven minutes, after which no significant events were identified in the diffraction pattern. The researchers concluded that the reaction takes place in two steps between 110–150 °C for both conventional and microwave heating, with no observable difference in the phase change, but some in the ongoing crystallisation of the product.

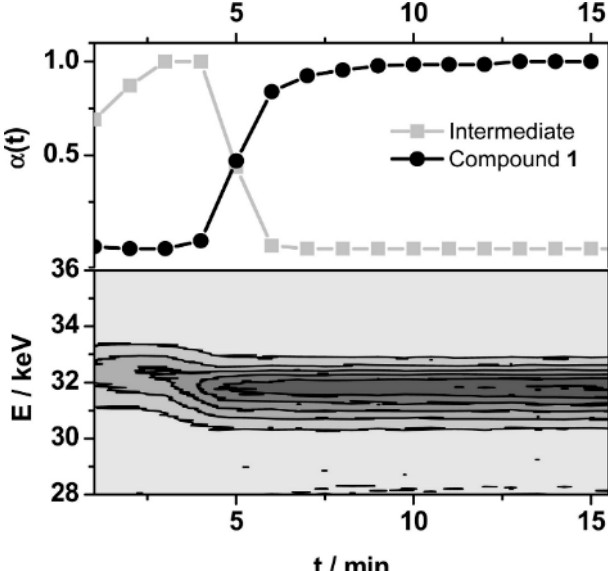

**Figure 18.** Surface energy-dispersive x-ray diffraction (EDXRD) plot of the transformation of the intermediate phase into $Ln(O_3P-C_4H_8-SO_3)(H_2O)$ (bottom) and reaction progress ($\alpha(t)$) for both phases under conventional heating at 150 °C (top). Reprinted with permission from reference. [95]. Copyright 2010, American Chemical Society.

The same group also carried out a temperature-dependent in situ EDXRD study for the formation and temperature-induced phase transition of previously described copper phosphonatosulfonates [96]. It was shown that at 90 °C, the formation of $[Cu_2(O_3PC_2H_4-SO_3)(OH)(H_2O)]\cdot 3H_2O$ is achieved after five minutes, but proceeds via a previously unknown phase, i.e., a tetrahydrate analogue of the

previous compound, which they were able to obtain as a phase pure product by quenching the reaction mixture within the first few minutes. Alternatively, increasing the temperature from 90 °C to 150 °C saw the formation of the monohydrate analogue after 12 min, with full phase transformation at 15 minu. This in situ EDXRD investigation allowed the researchers to propose the likely reaction pathway, which was shown in Figure 19.

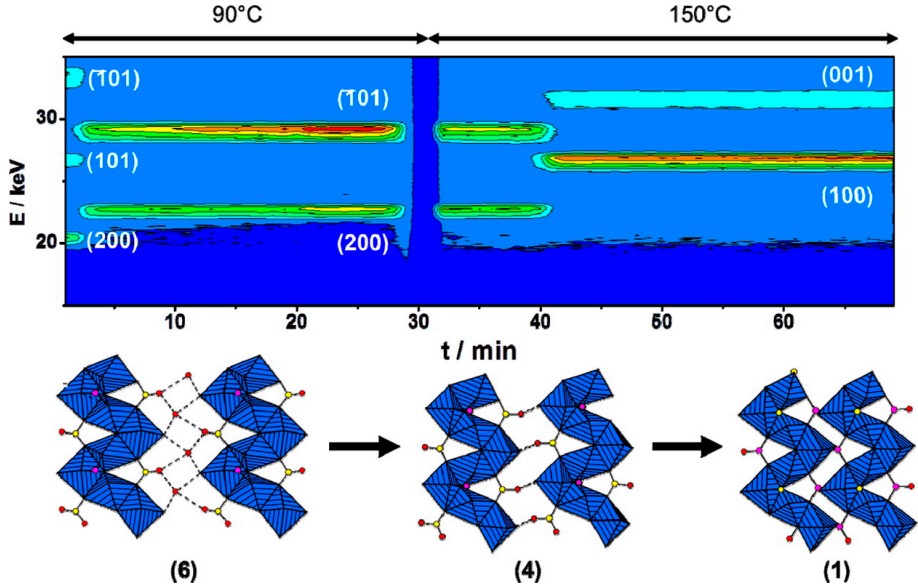

**Figure 19.** Illustration of the reaction pathway leading to the formation of [Cu$_2$(O$_3$PC$_2$H$_4$-SO$_3$) (OH)(H$_2$O)]·3H$_2$O **(1)** through intermediates Cu$_2$[(O$_3$P–C$_2$H$_4$–SO$_3$)(H$_2$O)$_2$(OH)]·4H$_2$O **(4)** and [Cu$_2$(O$_3$P–C$_2$H$_4$–SO$_3$)(H$_2$O)$_2$(OH)]·3H$_2$O **(6)**. The phase transformation at different temperatures is shown in the contour plot (top), and the correlated structural motifs of the copper oxygen chains are presented at the bottom. The gap in the contour plot at ~30 min is due to the stage at which the specimen was removed from the vessel. Reprinted with permission from reference. [96]. Copyright 2012, American Chemical Society.

In 2017, Heidenreich et al. [93] published a detailed overview of a novel reaction cell with integrated stirring and heating systems (up to 200 °C), which was named *SynRAC*, and used for in situ investigation of reactions under hydrothermal or solvothermal conditions using synchrotron radiation. The setup and design of the cell is discussed in detail in the paper: the main advantage of the *SynRAC* is that it is designed to be as similar as possible to a common laboratory reaction vessel, which allows for the extensive and reliable preliminary ex situ screening of reaction parameters before the actual in situ work, thus helping maximise the amount of information obtainable from the available synchrotron beamtime. A number of case studies are also presented to demonstrate the methods they use and the results that can be achieved. These do not include any MPs, but the *SynRAC* could easily find employment for this class of materials as well.

Section 3.2 has already suggested that in situ studies can be very informative for mechanochemical syntheses [56–59,97]. The two techniques used in each of the in situ studies reported so far were XRD and Raman spectroscopy, which allow the reaction to be monitored at both molecular and crystalline levels [97]. Traditionally, the vessel used in the milling/grinding process would be lined with an abrasion-resistant material, and would contain the grinding media, which is often constructed from steel, ceramic, and other suitable materials. However, for the in situ studies, the vessel needed to be adapted, which led to the grinding vessel to be made of Perspex. This allows radiation from both techniques to penetrate the vessel and reach the appropriate detector.

Combined XRD and Raman data on metal phenylphosphonates [57] has allowed the researchers to identify various phases throughout reactions. In 2015, Batzdorf et al. [97] described the in situ

investigation for the synthesis of cobalt(II) phenylphosphonates monohydrate (CoPhPO$_3$·H$_2$O), where equimolar amounts of cobalt(II) acetate and phenylphosphonic acid were ground together to form the product. The in situ XRD and Raman data confirmed that the product started forming after only 2.75 min of grinding, with completion of the reaction occurring at 27 min (Figure 20a,b). Different stages of the reaction were also identified, the main being where the product coexisted with intermediate phases.

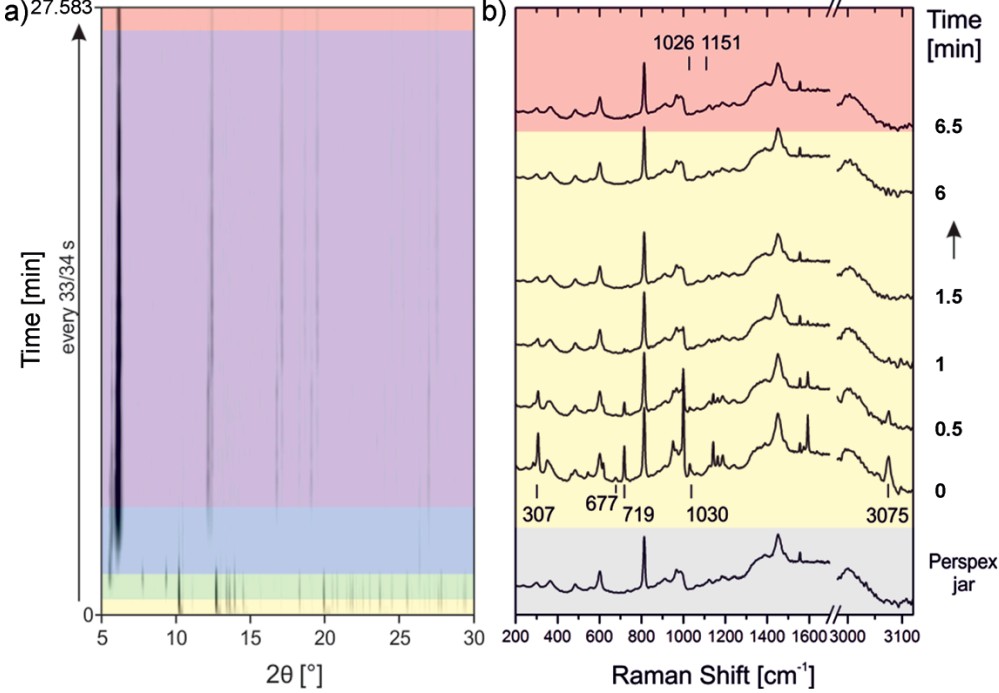

**Figure 20.** Synthesis process of Co(PhPO$_3$)·H$_2$O followed in situ by (**a**) synchrotron X-ray diffraction (XRD) and (**b**) Raman spectroscopy. The first measurement of the Raman plot (grey) is the Raman spectrum of the empty Perspex jar, indicating the modes arising from the sample holder. Colour code: yellow: reactants, green: reactants and intermediate phases, blue: reactants, intermediate phases and product, purple: intermediate phases and product, red: product. Adapted with permission from reference [97]. Copyright 2015, Wiley-VCH Verlag GmbH and Co.

An example where in situ analysis was able to provide unparalleled insight is the formation of Mn(HO$_3$PPh)$_2$(H$_2$OPPh)$_2$(H$_2$O)$_2$. XRD data indicated that there were five phases during the reaction [57] (Figure 21a,b). The initial reflections represent the starting materials (phase 1), and it is not until 30 s into the reaction that additional reflections indicate the appearance of a new phase (phase 2), which itself lasts approximately 30 s. At this point, strong reflections for the final product can be observed alongside those of Mn(O$_3$PC$_6$H$_5$)·H$_2$O (phase 3). The reflections for both starting materials disappear at 1.15 min, after which the Mn(O$_3$PC$_6$H$_5$)·H$_2$O reflections start to decrease, while those of the final product intensify (phase 4). Then, at six minutes, the reflections for anything other than the product reach a minimum and go through no further changes (phase 5). Raman data, after 30 s of milling, shows bands exclusively for the uncoordinated phenylphosphonic acid. After an additional 30 s, bands assigned to the coordinated phenylphosphonic acid start to appear, followed by the disappearance of the uncoordinated phosphonic acid band. The final change is the increase in intensity for the coordinated phenylphosphonic acid band.

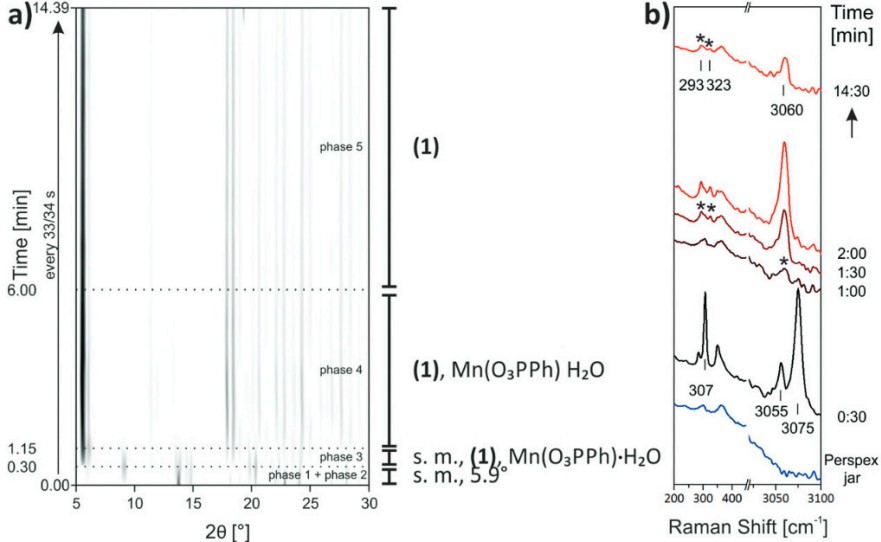

**Figure 21.** Two-dimensional (2D) plots of (**a**) synchrotron X-ray diffraction (XRD) data with a description of the detected compounds and (**b**) in situ Raman spectroscopy measurements monitoring the synthesis of $Mn(HO_3PPh)_2(H_2O_3PPh)_2(H_2O)_2$. Adapted with permission from reference. [57]. Copyright 2016, Royal Society of Chemistry.

Based on the aforementioned examples, it is clear that the use of in situ investigations can provide invaluable information on the mechanism of a reaction as well as the different crystalline phases appearing before the reaction completion. Furthermore, they present opportunities to optimise conditions to isolate specific new materials that might exist as intermediate phases for products with previously known structures.

## 4. Applications of Metal Phosphonates

### 4.1. Catalysis

Catalysis has played a key role in enabling the huge progress of the chemical industry in the 20th century [98]. Most of the large-scale industrial processes producing essential commodities, such as ammonia, sulfuric acid, nitric acid, and polyolefins, are made possible by the use of heterogeneous catalysts [99,100]. Typically employed catalysts in industrial settings consist of metal nanoparticles deposited on the surface of solid supports, such as metal oxides, carbon-based materials or zeolites [101,102]. With the advent of MOFs and the precise atomic control that they allow, it has also become possible to introduce active sites, which are most often coordinatively unsaturated metal atoms, in the porous framework [103,104]. Drawing inspiration from earlier work on the use of zirconium and titanium phosphates as solid acid catalysts [105], initial applications of MPs in catalysis were focused on, taking advantage of the Brønsted acidity of pendant sulfonic groups, which were successfully employed for cracking reactions in mild conditions [106]. This section presents recent examples of MPs employed both as supports for catalytically active species and as catalysts themselves.

A crucial feature of effective catalyst supports is a large surface to make the active species easily accessible to reactants. This can be achieved by the use of porous supports or by the exfoliation of layered materials to generate nanosheets consisting of single or few layers. A representative example of the latter strategy involves a mixed zirconium phosphate/phosphonate of formula $Zr_2(PO_4)H_5[(O_3PCH_2)_2NCH_2COO]_2 \cdot H_2O$ (Figure 22) containing N,N-bis[(phosphonomethyl)glycine] (glyphosine) as a ligand [107]. The interlayer region of this structure is characterised by the presence of both carboxylic and phosphonic acid groups, which make the layer surface highly polar and give rise to an extended network of hydrogen bonds. The presence of these groups make the compound very prone to ion exchange and the intercalation of small organic amines, such as propylamine, yielding

stable colloidal dispersions of hybrid nanosheets. The addition of palladium acetate to the dispersion led to the coordination of the metal to the non-coordinating carboxylic and phosphonic acid groups and enabled the deposition of palladium nanoparticles with size <2 nm onto the nanosheets with variable loadings, with the highest being reported at 19 wt. % [108]. The catalyst loaded with 15 wt. % of Pd was tested for the Suzuki–Miyaura coupling between phenylboronic acid and various aryl bromides, both in batch and continuous flow conditions, showing excellent performance and recyclability (Table 1 and Scheme 1). Importantly, the minimal leaching of Pd and no significant nanoparticle sintering were observed, suggesting the effectiveness of the support in binding the catalytically active species. The same catalyst was later successfully employed also for the hydrogenation in batch conditions of alkynes and nitroarenes [109] and for the Heck reaction [110], proving its wide applicability.

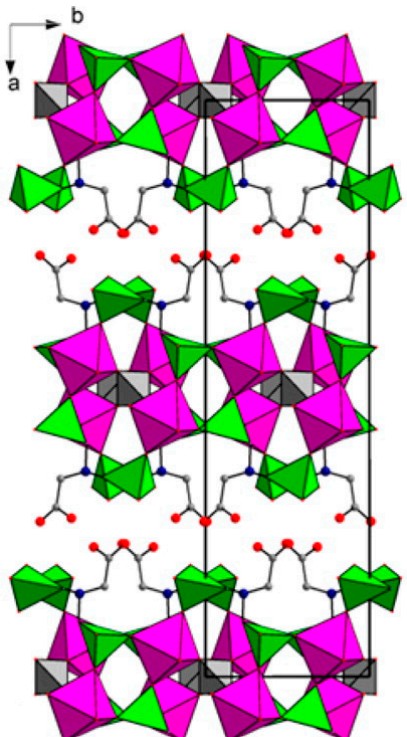

**Figure 22.** Crystal structure of $Zr_2(PO_4)H_5[(O_3PCH_2)_2NCH_2COO]_2 \cdot H_2O$ viewed along the *c*-axis. Colour code: zirconium: pink, phosphorus: green, oxygen: red, nitrogen: blue, carbon: grey. Reprinted with permission from reference. [107]. Copyright 2014, American Chemical Society.

**Scheme 1.** Representative Suzuki–Miyaura coupling catalysed by Pd nanoparticles supported on $Zr_2(PO_4)H_5(O_3PCH_2)_2NCH_2COO)_2 \cdot H_2O$ nanosheets (15 wt. % Pd). Reprinted with permission from ref. [108]. Copyright 2015, Royal Society of Chemistry.

**Table 1.** Results of catalytic tests for the Suzuki–Miyaura coupling displayed in Scheme 1. Reprinted with permission from ref. [108]. Copyright 2015, Royal Society of Chemistry.

| Entry | Aryl Bromide | R | *t* (h) | Yield (%) |
|---|---|---|---|---|
| 1 | **2a** (1st) | Me | 30 | 97 |
| 2 | **2a** (2nd) | Me | 30 | 97 |
| 3 | **2a** (3rd) | Me | 30 | 97 |
| 4 | **2b** | H | 30 | 96 |
| 5 | **2c** | CHO | 10 | 98 |
| 6 | **2d** | NO$_2$ | 10 | 98 |

More recently, a new strategy was adopted to obtain a zirconium phosphate–phosphonate decorated with very small gold nanoparticles on its surface [111]. The procedure consisted of different steps: first, a nanosized zirconium phosphate was obtained using a synthesis from gel in propanol [112]. Then, due to the small dimensions and high defectivity of the nanocrystals, this compound could be functionalised with amino groups through a topotactic anion exchange reaction of phosphates with incoming aminoethylphosphonate groups, and a compound with formula Zr(PO$_4$)$_{1.28}$(O$_3$PC$_2$H$_4$NH$_2$)$_{0.72}$ (hereafter ZP-AEP) was prepared (Figure 23). Finally, gold nanoparticles with dimensions less than 10 nm were produced by reduction, with NaBH$_4$, of AuCl$^{4-}$ ions chemisobted on the surface of nanocrystals (Figure 24).

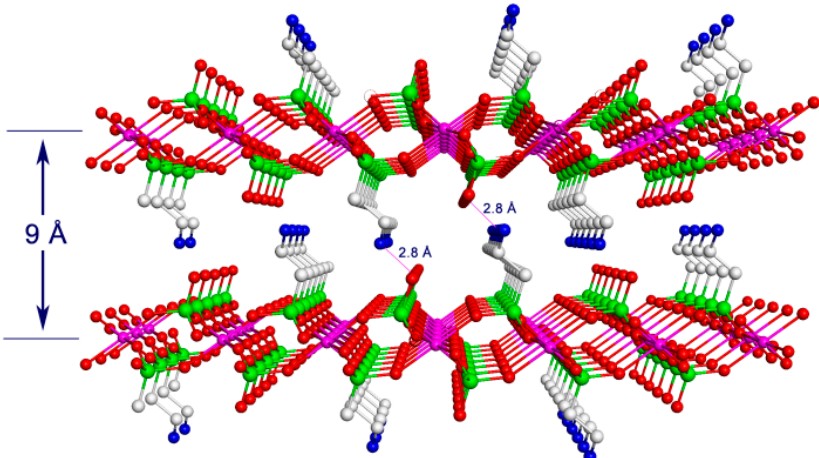

**Figure 23.** Schematic representation of a mixed α-Zr phosphate aminoethylphosphonate. Adapted with permission from reference [111]. Copyright 2019, Royal Society of Chemistry.

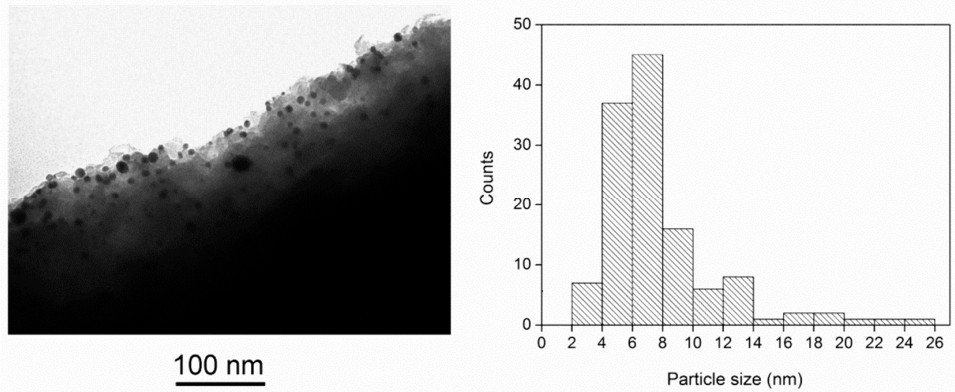

**Figure 24.** Transmission electron micrograph of Au@ZP-AEP (**left**) and Au particle size distribution (**right**). Adapted with permission from reference [111]. Copyright 2019, Royal Society of Chemistry.

A sample obtained with this procedure, and containing 1 wt. % Au, was found to be an efficient catalyst for the chemoselective reduction of a series of nitroarenes (Scheme 2); furthermore, the authors showed that it was possible to efficiently switch the reaction product between the corresponding azoxyarenes or anilines, simply by changing the solvent (96% EtOH or absolute EtOH, respectively). The recovery and reuse of the catalyst was also very efficient, as shown in Table 2.

**Scheme 2.** Representative reduction of nitroarenes using Au@ZP–AEP as the catalyst and $NaBH_4$ as the reducing agent.

**Table 2.** Representative results of Au@ZP–AEP catalyst in the switchable preparation of an azoxy derivative (**2a**) and methoxyaniline (**4a**) and its recovery and reuse. Reprinted with permission from reference. [111]. Copyright 2019, Royal Society of Chemistry.

| Entry [a] | Run | Medium | $t$ (h) | $C$ [b] (%) | 2 : 3 : 4 Ratio [b] |
|---|---|---|---|---|---|
| 1 | Run 1 | 96% EtOH | 3 | 98 | 96 : 4 : 0 |
| 2 | Run 2 | 96% EtOH | 3 | 96 | 97 : 3 : 0 |
| 3 | Run 3 | 96% EtOH | 3 | 94 | 97 : 3 : 0 |
| 4 | Run 4 | 96% EtOH | 3 | 93 | 96 : 4 : 0 |
| 5 | Run 5 | 96% EtOH | 3 | 87 | 98 : 2 : 0 |
| 6 | Run 1 | $EtOH_{abs}$ | 2 | >99 | 0 : 0 : 100 |
| 7 | Run 2 | $EtOH_{abs}$ | 2 | >99 | 0 : 0 : 100 |
| 8 | Run 3 | $EtOH_{abs}$ | 2 | >99 | 0 : 0 : 100 |
| 9 | Run 4 | $EtOH_{abs}$ | 2 | >99 | 0 : 0 : 100 |
| 10 | Run 5 | $EtOH_{abs}$ | 2 | >99 | 0 : 0 : 100 |

[a] Reaction conditions: **1** (0.1 mmol), Au@ZP–AEP (1 mol%), $NaBH_4$ (6 equivalents), reaction medium: 96% EtOH or $EtOH_{abs}$ (1.8 mL), at 30 °C. [b] Conversion and product ratios determined by gas liquid chromatography and $^1$H-NMR analyses.

A notable feature of both STA-12(Ni) and STA-12(Co) is their ability to be activated in order to create coordinatively unsaturated metal sites [113]. This occurs due to the loss of chemisorbed water at around 85–109 °C. Given the high density of these Lewis acidic metal sites, the STA-12 family of MPs has been investigated for a range of catalytic reactions. STA-12(Co) has been particularly notable for the catalysis of aerobic epoxidation of olefins, which are typically catalysed by Co-doped zeolites [114]. The issue with zeolitic materials in this case is that in order to achieve the required isolated Co species, the loading of Co in the zeolite must be relatively low, which means that larger quantities of the catalyst will be required. This is where MOFs have shown promise over other materials, in that they are often, by nature, high metal-containing materials. The catalytic activity of STA-12(Co) was found to be comparable to the zeolite catalysts, obtaining similar results even when reducing the amount of catalyst by two orders of magnitude. It was also found that the substrate used could have a great effect on the selectivity. With styrene, the selectivity to styrene oxide was low due to substrate oligomerization. Conversely, for (E)-stilbene, and (Z)-stilbene, the selectivities for the relative oxides were shown to be

between 80–90%, with no considerable oligomerization of the substrate (Figure 25). It was also shown that the catalyst could be recycled, with little compromise on the activity. STA-12(Ni), similar to its Co analogue, has also shown reasonable catalytic activity. Both materials also showed little or no change in crystallinity when reused, with only a 1% difference in product formation after three cycles.

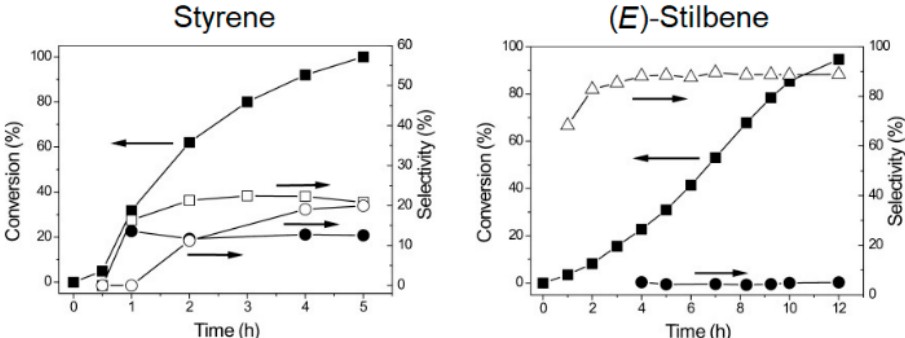

**Figure 25.** The epoxidation reaction of styrene (**left**) and (*E*)-stilbene (**right**) with the Co-based metal–organic framework material STA-12(Co). The graph on the left shows styrene conversion (■), as well as the selectivity for styrene oxide (□), benzaldehyde (●), and benzoic acid (○). The graph on the right shows (*E*)-stilbene conversion (■), as well as the selectivity for *trans*-stilbene oxide (Δ) and benzaldehyde (●). Adapted with permission from reference. [114]. Copyright 2012, Wiley-VCH Verlag GmbH & Co.

The combination of the high stability and structural versatility of MPs can be features promoting their employment in heterogeneous catalysis. As for MOFs, the presence of organic groups prevents their employment in high-temperature processes similar to those that are typical of the heavy petrochemical industry. However, they could potentially find application in the production of fine chemicals and catalytic processes of importance for sustainability issues, such as the conversion of $CO_2$ into value-added feedstocks, methane oxidation to methanol, electrocatalytic or photocatalytic hydrogen production, and water oxidation, which involve less harsh conditions.

*4.2. Gas Sorption and Separation*

Microporous MPs have the potential to be ideal candidates for gas sorption due to their high stability, especially in humid conditions, which often exceeds that displayed by carboxylate-based MOFs [12]. Unfortunately, as already discussed in Section 3.3, it has proved difficult for researchers to obtain porous MPs, since they are prone to forming densely packed layered structures, and only a small number have been reported to date [14,73]. Microporous MPs usually feature relatively low surface areas and pore volumes, which limits their scope for gas storage. However, most of them display channel-like pores, with diameters often below 10 Å, which can in principle provide favourable interaction between specific adsorbate species and the sorbent surface, which is a key requirement for the efficient separation of gaseous mixtures [115,116].

One of the most promising porous MPs identified for gas separation so far is MIL-91(Al/Ti), which features channels 3.5 × 3.5 Å in size [67,117]. On carrying out the adsorption experiments on $CO_2$, $CH_4$, and $N_2$ at 303 K, a strong affinity of MIL-91(Al) for $CO_2$ was observed (Figure 26) [69]. For both $CH_4$ and $N_2$, the isotherms showed no significant uptake of the gases, and failed to show any saturation plateau, even at ~50 bar. On the other hand, the isotherm for $CO_2$ showed significant uptake, with the majority of the gas being adsorbed at pressure below 1 bar, and complete saturation reached at ~15 bar. Microcalorimetric experiments showed that the enthalpy of $CO_2$ adsorption is constant at about 40 kJ·mol$^{-1}$ up to a loading of 4 mmol·g$^{-1}$, suggesting a strong physisorptive character, which is likely induced by the close interaction of $CO_2$ with the surrounding pore walls.

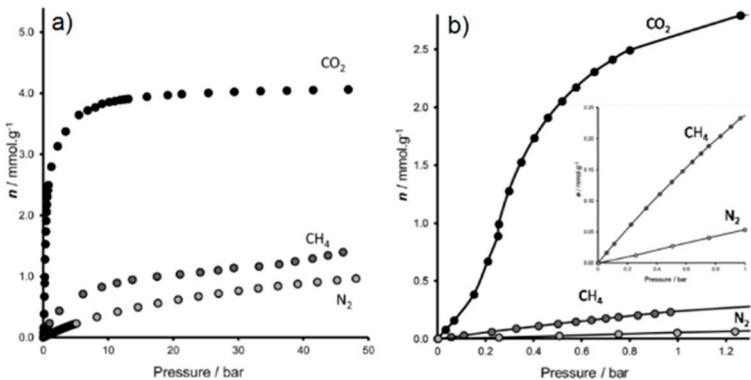

**Figure 26.** $CO_2$, $CH_4$, and $N_2$ adsorption isotherms measured at 303 K on MIL-91(Al) up to 50 bar (**left**) and up to 1 bar (**right**). Adapted with permission from reference. [69]. Copyright 2015, American Chemical Society.

Furthermore, an inflection on the $CO_2$ adsorption isotherm was identified at low pressure (Figure 26), which is due to a cooperative phenomenon involving the twisting of the ligand when a threshold $CO_2$ pressure is reached (Figure 27). This "phase-change" behaviour is especially interesting for its potential to combine the high working capacity and low energy penalty for the regeneration of the sorbent. The MIL-91(Ti) analogue showed a similar preference for $CO_2$ over the other gases, with a steep uptake at low pressure and saturation exceeding 4.0 mmol·$g^{-1}$ [68]. However, unlike the Al analogue, no flexibility in the structure was observed, which was exemplified by the lack of inflection or S-character in the isotherm. Overall, MIL-91(Ti) was identified to be a viable material for $CO_2$ capture, owing to increased thermal stability when compared to other MOFs, its ability to selectively adsorb $CO_2$ over other gases, and the possibility to be produced in quantities beyond the laboratory scale.

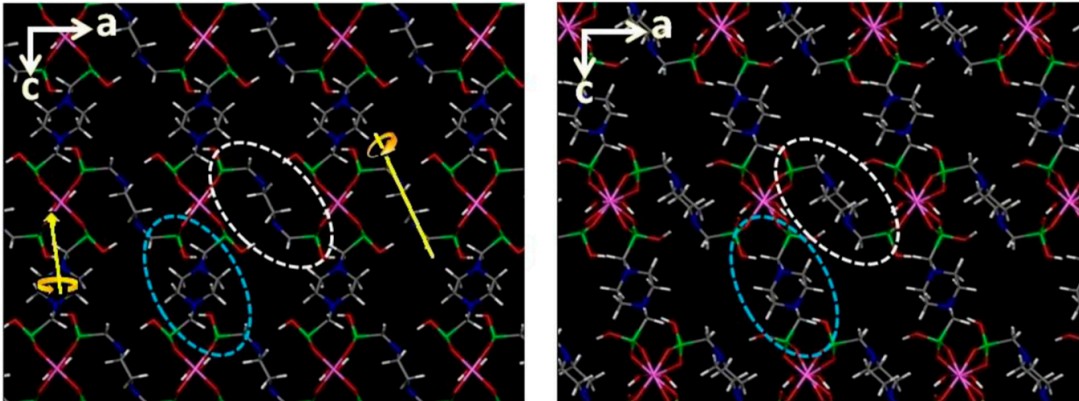

**Figure 27.** Crystal structure of MIL-91(Al) before (**left**) and after (**right**) the adsorption of $CO_2$. The rotation of crystallographically independent linker units is highlighted within the white and blue circles. Reprinted with permission from reference. [69]. Copyright 2015, American Chemical Society.

The gas separation properties of the STA-12 framework were also thoroughly investigated for the analogues containing Ni, Co, and Mg [71]. IR experiments with CO and $CO_2$ as probes showed that despite the presence of coordinatively unsaturated metal sites, none of the STA-12 frameworks display a strong interaction of the adsorbates with these sites. By the adsorption of $N_2$ at 77 K and $CO_2$ at 195 K, it was found that STA-12(Ni) is the most porous compound of the series, and further characterisation demonstrated that it has a high selectivity for $CO_2$ over $CH_4$ at ambient temperature [118]. The separation performance of STA-12-Ni was compared with that of the CPO-27-M (M = Ni, Co, and Zn), which is a carboxylate-based MOF with similar pore structure and a high density of open metal sites exposed on the channels [119]. The MOFs were tested for both binary

50:50 $CO_2/CH_4$ mixtures and ternary 70:15:15 $CO_2/CO/CH_4$ mixtures by means of single gas isotherms and breakthrough analysis (Figure 28a–d). The CPO-27 frameworks outperformed STA-12 for the separation of the binary mixture, thanks to the strong interaction of $CO_2$ with the open metal sites. On the other hand, when exposed to the ternary mixture, CPO-27-Ni and CPO-27-Co displayed preferential adsorption of CO, whereas STA-12(Ni) and CPO-27-Zn maintained their selectivity for $CO_2$. STA-12(Ni) crystallises as submicron-size particles suitable for use as the stationary phase in a porous layer open tubular (PLOT) capillary column, which was used to separate alkanes according to boiling point, giving promising separation performance, even without optimization [71]. The stability and versatility of MOF structures suggest that they can find specialist application in this field.

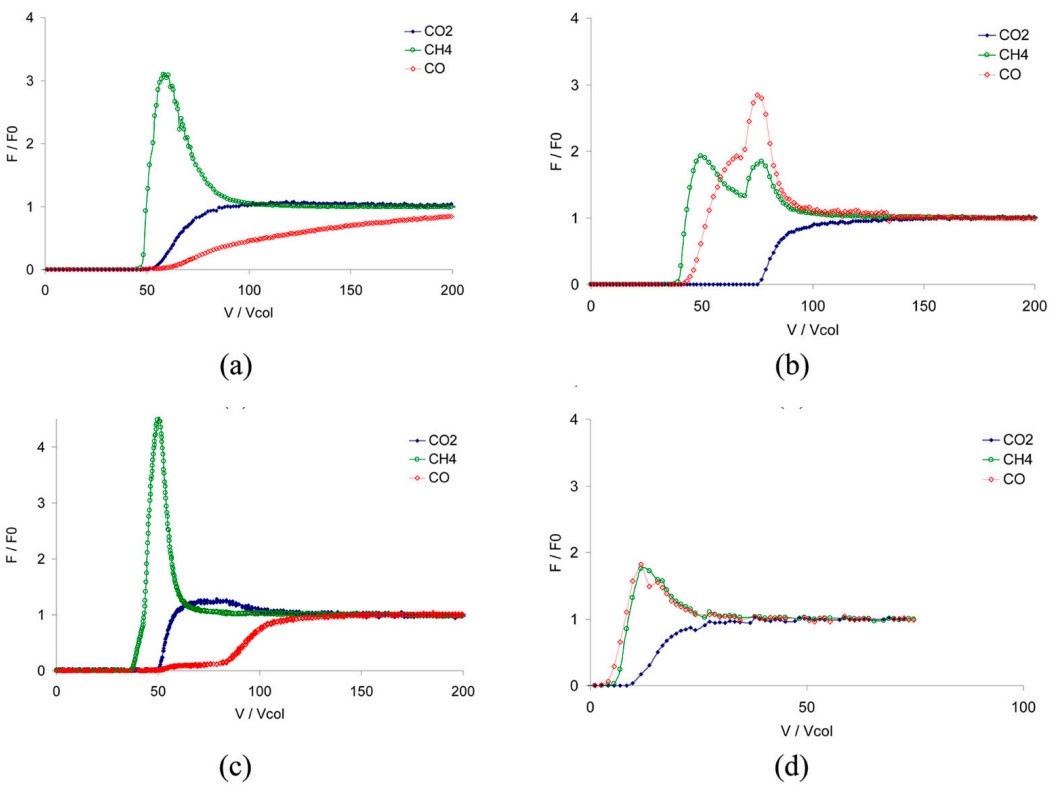

**Figure 28.** Breakthrough curves of the $CO_2/CH_4/CO$ (70/15/15) mixture on (**a**) CPO-27-Co, (**b**) CPO-27-Zn, (**c**) CPO-27-Ni, and (**d**) STA-12-Ni at 303 K and 5 bar. Reprinted with permission from reference. [71]. Copyright 2011, Elsevier Inc.

Given the recent progresses made in synthesising porous MPs, more intensive investigation of their gas sorption and separation properties is likely to take place in the near future. Thanks to the enormous success of MOFs, gas sorption analysis has become a mainstream technique that can be readily employed for systematically assessing the potential of microporous MPs for a range of gas separations, especially those where challenging conditions can limit the applicability of carboxylate-based MOFs.

## 4.3. Electrochemical Devices

While global energy demand grows year on year, with a projected 30% increase by the year 2040 [120] there has been a lot of interest in finding new, more efficient, and cleaner ways of powering our future. In this context, electrochemical devices such as rechargeable batteries and fuel cells are becoming increasingly important for energy storage and conversion, attracting considerable research efforts [121,122]. Technically, a battery and a fuel cell are both electrochemical cells, consisting of an anodic and a cathodic compartment, which are connected through an electrical circuit that allows the exchange of electrons, while an electrolyte ensures the mobility of positive charge carriers (Figure 29) [123]. The main difference between the two devices is that rechargeable batteries can

reversibly store electricity in the form of chemical energy and release it on demand, whereas fuel cells convert the chemical energy contained in a fuel (typically hydrogen) into electricity. The interest in maximising the performance of these devices is currently a major drive in developing new materials that can serve as either electrodes or electrolyte components [123].

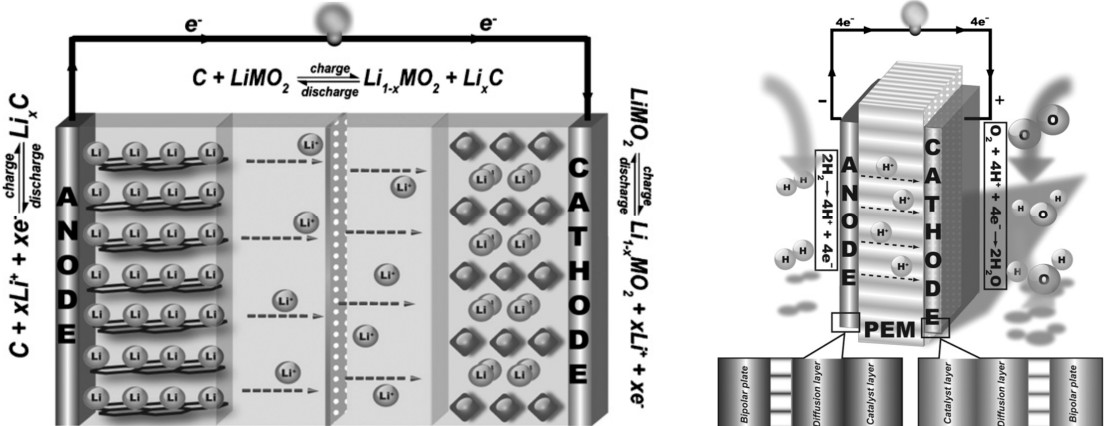

**Figure 29.** Schematic representations of a rechargeable lithium ion (Li-ion) battery (**left**) and a proton exchange membrane fuel cell (**right**). Adapted with permission from ref [123]. Copyright 2009, Elsevier Inc.

4.3.1. Solid-State Proton Conductors for Fuel Cells

The positive charge carriers in fuel cells are protons, which need to be efficiently transported via the electrolyte from the anode, where $H_2$ is oxidised, to the cathode, where they combine with oxygen to form water, in order to maximise the power generation. This makes the electrolyte a crucial component in determining the efficiency of a fuel cell, driving research into materials displaying high proton conductivity [124,125]. The most common type of fuel cell is the proton exchange membrane fuel cells (PEMFC), where the electrolyte is typically a proton conductive polymer, such as Nafion, polybenzimidazole, and sulfonated polyether-ether ketones. In spite of their very high conductivity, these polymers often suffer from limited thermal and mechanical stability, which affects their long-term performance. As early as the 1990s, layered zirconium phosphonates were proposed as alternative proton conduction materials [126,127], owing to the ability to functionalise them with groups of variable acid strength. In particular, zirconium sulfophenylphosphonate was found to have excellent proton conductivity in humid conditions, reaching values as high as $1.6 \times 10^{-2}$ S·cm$^{-1}$. The application of MPs as proton conductors has recently been extensively reviewed [6,128]; therefore, we will only present some selected examples.

A recent example of a metal phosphonate with good proton conduction is that of La(H$_5$DTMP)· 7H$_2$O, which is based on the hexamethylenediamine-*N,N,N′,N′-tetrakis*(methylenephosphonic acid) linker (Figure 30a) [129]. Structural analysis of the compound showed that it is a three-dimensional (3D) framework featuring narrow one-dimensional (1D) channels where seven water molecules per formula unit are accommodated, forming a network of hydrogen bonds involving non-coordinated P–O groups that extend throughout the channels (Figure 30b). This allows for efficient proton conduction to take place, as proved by the conductivity value of $8 \times 10^{-3}$ S cm$^{-1}$ at 25 °C and 98% relative humidity, as measured by impedance analysis. The activation energy was found to be 0.23 eV, which is typical of a Grotthuss-type mechanism, where well-ordered water molecules play a key role in enabling efficient proton shuttling. Other MPs with similar "water channels" were subsequently reported [130–133], which displayed good proton conduction and activation energies consistent with the Grotthuss mechanism.

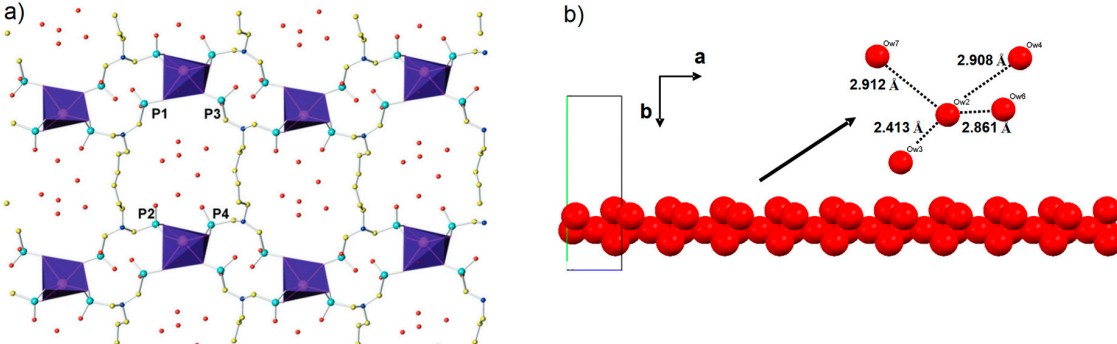

**Figure 30.** (**a**) Crystal structure of La(H$_5$DTMP)·7H$_2$O and (**b**) the extended network of hydrogen-bonded water molecules along the *a*-axis, along with a detail of the hydrogen bonding distances. Colour code: lanthanum: purple, phosphorus: light blue, oxygen: red, nitrogen: dark blue, carbon: yellow. Adapted with permission from reference [129]. Copyright 2012, Royal Society of Chemistry.

These results highlight the importance of having well-defined pathways for the protons to hop between charge carriers to maximise conductivity. Clearly, knowledge of the crystal structure is essential to be able to identify such pathways in the first place. It is also worth emphasising that using polyphosphonic acids as linkers is crucial, since this increases the chances of having free P–OH groups that can facilitate the formation of extended hydrogen bond networks with water molecules. These characteristics can be enhanced by post-synthesis modifications, for instance by the adsorption of molecules, which contribute to generate robust proton transfer pathways [130].

### 4.3.2. Electrodes for Rechargeable Batteries

Different from fuel cells, the efficiency of a rechargeable battery not only depends on the mobility of the charge carriers, but heavily relies on the ability of the electrodes to reversibly store and release these charge carriers during charging and discharging. Typical charge carriers are monovalent alkaline ions, with lithium element presently being the most employed, owing to its small size and low weight, which ensure high charge density. In order to maximise the performance, it is important that both electrodes possess high specific capacity and long-term stability. The state of the art of the electrodes for lithium ion (Li-ion) batteries includes graphite as the anodic material (which is present in the negative electrode, theoretical specific capacity = 372 mAh·g$^{-1}$) and LiCoO$_2$ as the cathodic material (which is present in the positive electrode, theoretical specific capacity = 137 mAh·g$^{-1}$). They are both layered compounds where Li-ions are inserted and extracted by intercalation and deintercalation processes, respectively. The major limitation of the energy density of present Li-ion batteries is the relatively low specific capacity of the cathode. In addition, the latter is slowly decreasing during cycling due to the material instability (Co dissolution in the electrolyte) and other mechanisms.

Recently, a MP based on Fe(II) and methylenediphosphonic acid, of formula Li$_{1.4}$Fe$_{6.8}$ [CH$_2$(PO$_3$)$_2$]$_3$[CH$_2$(PO$_3$)(PO$_3$H)]·4H$_2$O, was extensively studied for its potential as a positive electrode for Li-ion batteries (Figure 31a,b) [134]. Structural characterisation was carried out combining synchrotron PXRD and neutron diffraction data, finding that the compound has a very similar structure to a previously reported Co methylenediphosphonate [30]. Three different crystallographic sites were identified for Fe(II) ions: two of them are octahedrally coordinated, while the other one is tetrahedrally coordinated. The compound was prepared in the presence of Li, which was retained within the crystal structure as an extra-framework species that was accommodated in the small 1D channels, thus rendering a pre-lithiated electrode. Electrochemical testing showed a specific charge of 85 mAh·g$^{-1}$ until the 60th cycle, corresponding to 50% of the theoretical value of 168 mAh·g$^{-1}$. The origin of this lower value was attributed to part of the Fe ions not undergoing any redox process, as evidenced by in situ near-edge X-ray absorption spectroscopy. Ex situ XRD analysis after the first electrochemical

cycle showed that there was no significant loss of crystallinity, suggesting that activation did not proceed via decomposition of the compound. Successive studies focused on testing the same type of framework as a negative electrode, using both Fe and Co as the metal component [135], and combining methylenediphosphonic acid with Ni, which leads to the formation of different crystal structures [136]. In all cases, it was found that the structure irreversibly amorphises after the first cycle due to a conversion reaction mechanism involving the extrusion of the transition metal(II) as nanoparticles.

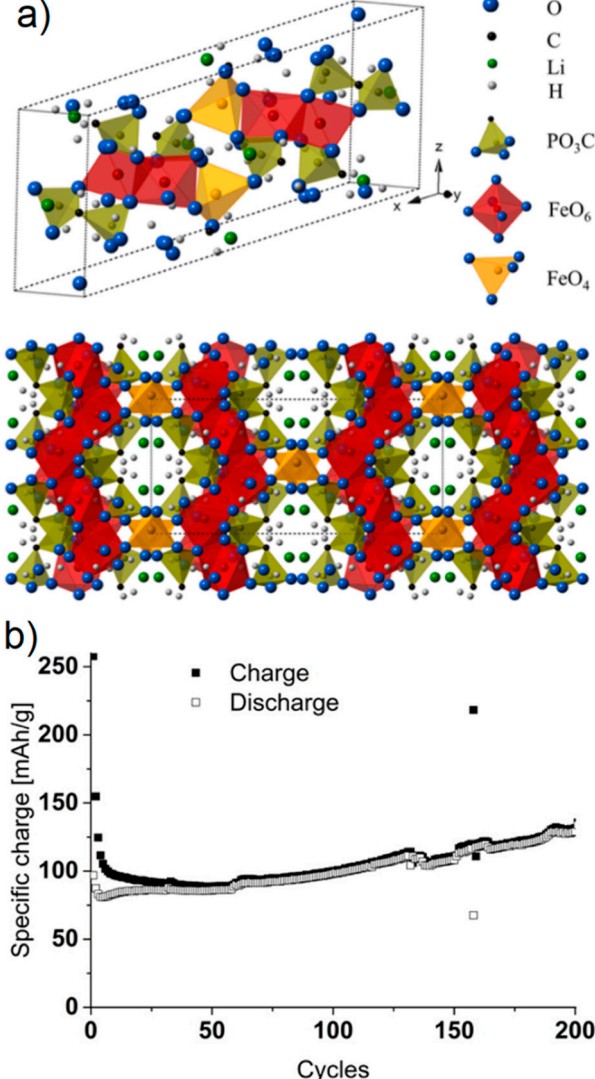

**Figure 31.** (**a**) Illustration of the monoclinic unit cell of $Li_{1.4}Fe_{6.8}[CH_2(PO_3)_2]_3[CH_2(PO_3)(PO_3H)]\cdot4H_2O$ and the expanded crystal structure viewed along the *c*-axis. Colour code: octahedral iron: red, tetrahedral iron: orange, phosphorus: yellow, oxygen: blue, carbon: black, water: grey, and lithium: green. (**b**) Specific charge vs. cycle number plot of $Li_{1.4}Fe_{6.8}[CH_2(PO_3)_2]_3[CH_2(PO_3)(PO_3H)]\cdot4H_2O$ cycled at 20 mA/g with a 1-h potentiostatic step after each half-cycle. Adapted with permission from reference [134]. Copyright 2015, American Chemical Society.

The compounds discussed above represent some of the first reported examples of MPs being applied as electrode materials for rechargeable batteries. There is a growing interest in employing hybrid organic–inorganic materials for this type of application, mainly because of the possibility to tailor their structure, something that is not easily done when dealing with purely inorganic materials. Given the structural versatility of MPs and their high stability, there is great potential in this growing area of research.

### 4.4. Drug Delivery

Phosphonic acids (most often in their sodium phosphonate form) have featured in the treatment of bone-related conditions, e.g., osteoporosis and Paget's disease, for more than 30 years. Specifically, bisphosphonates (BPs) have been used as stable analogues of pyrophosphate, in which the P–O–P bonds are replaced with P–C–P bonds, making them more stable and less prone to enzymatic attack [137]. The way in which BPs can help in the treatment of bone-related conditions is through selective adsorption to the bone surface, leading to the interruption of bone resorption, whereby osteoclasts are prevented from resorbing bone tissue. There are two classes of BPs that can be used here: non-N-containing compounds, such as etidronate and clodronate, and N-containing compounds, such as pamidronate and alendronate. Perhaps one of the main drawbacks for the use of BPs for treating conditions such as osteoporosis is the limited bioavailability when delivered orally [138]. This means that in order to obtain the desired therapeutic effect, it is often the case that the drug dosage is increased. While this appears to overcome the bioavailability issue, it can also lead to an increase in other known side effects, i.e., hypocalcaemia, oesophageal cancer, and atrial fibrillation, etc. In order to overcome this issue, delayed release (DR) administration and other methods have been developed, helping to negate the need to increase the dosage. An example of one of the DR drugs is risedronate 35 mg (DR). Unfortunately, these drugs still suffer in that they require long periods of fasting due to the complexing that can occur with food in the stomach.

In a paper published in 2019, Papathanasiou et al. explored various routes for the controlled release of bisphosphonates, one of which was based on self-sacrificial MOFs [139]. Using etidronic acid (ETID, one of the "early" BPs) as the linker, the researchers synthesised Ca–ETID and presented it in tablet form, comparing it to a control containing "free" ETID. The structure of Ca–ETID exists as a 1D chain, where each $Ca^{2+}$ centre is coordinated to three ETID linkers, the –OH group, and water (Figure 32). Then, these single chains form 'double chains' held together by a network of hydrogen bonds between ETID linkers. The release rates of the control and Ca–ETID tablets upon soaking in an aqueous solution at pH 1.3 (representative of the environment in the stomach) were determined by NMR spectroscopy, finding that the control showed 100% release after just 10 h, whereas Ca-ETID showed only 30% release in the same time period, reaching a maximum of 90% after 150 h (Figure 32). The slower release rate of Ca–ETID tablets was ascribed to the slow rate of hydrolysis of the metal–phosphonate coordination bonds in the solvent medium.

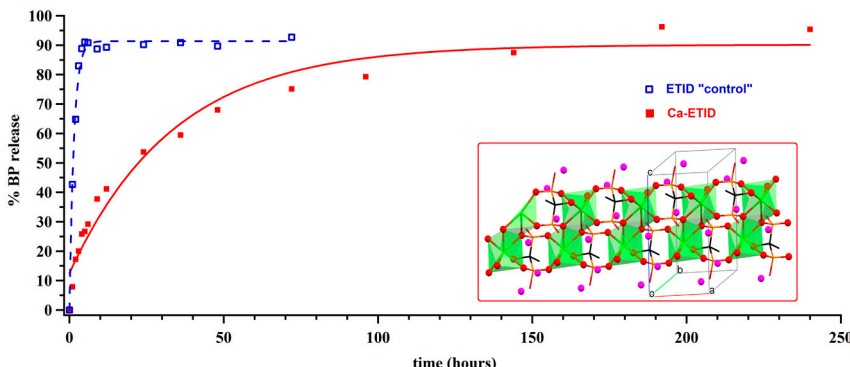

**Figure 32.** Etidronic acid (ETID) release versus time for the control tablet (dark blue circles) and the Ca–ETID tablet (light blue triangles). The crystal structure of Ca–ETID is also displayed in the inset. Colour code: calcium: green, phosphorus: yellow, oxygen: red, carbon: black, lattice water: pink. Adapted with permission from reference [139]. Copyright 2018, Walter de Gruyter GmbH and Co. KG.

These promising results are part of ongoing work that aims at systematically investigating the preparation and structural characterisation of a series of MPs based on various BP and biocompatible metals (e.g., Mg, Ca, Sr, and Ba), as well as their performance as matrices for controlled drug delivery. The final goal is to identify clear structure/activity relationships between specific structural features,

such as the metal–phosphonate bond strength, the metal ion radius, the existence of supramolecular interactions, and the drug delivery efficacy.

## 5. Outlook

Forty years after its inception, metal phosphonate chemistry has come a long way, but the breadth of topics discussed during the First European Workshop on Metal Phosphonate Chemistry, and reviewed here, shows that the field is still very lively and moving in new directions. From the synthesis standpoint, we have highlighted the potential of high-throughput methods in accelerating the discovery of new materials, the ability of mechanochemistry to provide access to structures not obtainable through conventional synthesis, and the intense efforts in developing permanently porous materials, taking inspiration from MOF chemistry, including the first demonstration of the suitability of phosphinic acids to serve as alternative building blocks for highly stable MOFs. On the characterisation side, the main advances discussed relate to the development of methods for structural solutions of microcrystallne and nanocrystalline compounds using electron diffraction, which might help overcome what is probably the longest standing challenge in the field. Furthermore, the employment of in situ methods can identify the different stages of the crystallisation process, which is knowledge that can be crucial to designing better synthetic processes. Looking at applications, the stability and structural versatility of MPs makes them suitable for employment in several fields: in catalysis, they can either serve as effective supports for catalytic nanoparticles or display intrinsic catalytic properties; the development of microporous phosphonate-based frameworks is also promoting investigation towards their employment in gas separations, as an alternative to other MOF classes; the ever-growing interest in developing new electrochemical devices is driving research aimed at evaluating their potential as both solid-state proton conductors and redox active materials; finally, the complexation of bisphosphonic drugs with alkaline earth metals shows promise as an effective strategy for controlled drug delivery. Our hope is that the First European Workshop on Metal Phosphonates Chemistry will serve as a sort of nucleation point for promoting the growth of a larger network of collaborations within the field, both at the European level and beyond, which would be crucial in order to make the next 40 years of metal phosphonates chemistry as productive and successful as the past 40 have been. A second edition of the workshop is taking place in Berlin in September 2019, demonstrating that there is indeed a strong interest within the community to keep the conversation open and look ahead to the future.

**Author Contributions:** Conceptualization, S.J.I.S. and M.T.; writing—original draft preparation, S.J.I.S. and M.T.; writing—review and editing, all authors.

**Funding:** M.T is supported by funding from the European Union's Horizon 2020 research and innovation program under the Marie Skłodowska-Curie grant agreement No 663830.

**Conflicts of Interest:** The authors declare no conflict of interest.

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
