# Peer review of "New Directions in Metal Phosphonate and Phosphinate Chemistry"

_crystals, doi:10.3390/cryst9050270_

Round 1

Reviewer 1 Report

Paper entitled "New Directions in Metal Phosphonate and Phosphinate Chemistry" is a 35-page long perspective article written as a summary (or: an outcome) of the 1st European Workshop on Metal Phosphonates Chemistry that took place in Swansea at the end of last year. It is written by specialists that are pioneers in various challenging areas of metal phosphonate chemistry: (electron, X-ray) diffraction, open frameworks, HT synthesis, proton conduction, in situ charaterization and many, many more.

As such, the paper is very comprehensive although covers so many research areas on 35 pages. It is expected to gain broad attention and will surely be highly citable item that will drive IF of Crystals journal to higher values.

Honestly speaking, as a reviewer I have completely no reservations to this article from the merithoric side and I recommend "as is" publication. The only things that could be polished are:

-  I dont see in the abstract nor in keywords section a phrase "proton conduction"; I think it would be beneficial to add this phrase into one of those search engine-indexed areas

- Some figures have really low resolution / high noise, see e.g. Fig. 5, Fig. 11. I am not sure whether this is the effect of low-res screenshots from previous papers or just the result of PDF compression. 

Author Response

“Paper entitled "New Directions in Metal Phosphonate and Phosphinate Chemistry" is a 35-page long perspective article written as a summary (or: an outcome) of the 1st European Workshop on Metal Phosphonates Chemistry that took place in Swansea at the end of last year. It is written by specialists that are pioneers in various challenging areas of metal phosphonate chemistry: (electron, X-ray) diffraction, open frameworks, HT synthesis, proton conduction, in situ charaterization and many, many more.

 As such, the paper is very comprehensive although covers so many research areas on 35 pages. It is expected to gain broad attention and will surely be highly citable item that will drive IF of Crystals journal to higher values.

 Honestly speaking, as a reviewer I have completely no reservations to this article from the merithoric side and I recommend "as is" publication.

Response: we warmly thank the Reviewer for the positive evaluation of our manuscript.

The only things that could be polished are:

-  I dont see in the abstract nor in keywords section a phrase "proton conduction"; I think it would be beneficial to add this phrase into one of those search engine-indexed areas

Response: we thank the Reviewer for this suggestion, we have added “proton conduction” as one of the keywords.

 - Some figures have really low resolution / high noise, see e.g. Fig. 5, Fig. 11. I am not sure whether this is the effect of low-res screenshots from previous papers or just the result of PDF compression.

Response: we thank the Reviewer for pointing this issue out. We have made a new Figure 5 from scratch and it now contains high-resolution structural pictures. As for figure 11, we have done our best to improve the quality, but we are limited by the resolution of the figure in the original manuscript.

Reviewer 2 Report

In this perspective, the authors provide a clear overview of the recent important developments in the synthesis, characterization, and applications of metal phosphonate and phosphinates, and of their impact on the future of the field. This is a valuable work, especially for young researchers entering the field, and therefore I recommend acceptance of this manuscript.

Author Response

In this perspective, the authors provide a clear overview of the recent important developments in the synthesis, characterization, and applications of metal phosphonate and phosphinates, and of their impact on the future of the field. This is a valuable work, especially for young researchers entering the field, and therefore I recommend acceptance of this manuscript.

Response: we warmly thank the Reviewer for the positive evaluation of our manuscript.